# Stella modulates transcriptional and endogenous retrovirus programs during maternal-to-zygotic transition

Yun Huang[1,2†], Jong Kyoung Kim[3,4†], Dang Vinh Do[1,2], Caroline Lee[1,2], Christopher A Penfold[1,2], Jan J Zylicz[1,2], John C Marioni[3,5,6], Jamie A Hackett[1,2,7*], M Azim Surani[1,2*]

[1]Wellcome Trust/Cancer Research United Kingdom Gurdon Institute, University of Cambridge, Cambridge, United Kingdom; [2]Department of Physiology, Development and Neuroscience, University of Cambridge, Cambridge, United Kingdom; [3]European Molecular Biology Laboratory, European Bioinformatics Institute, Cambridge, United Kingdom; [4]Department of New Biology, Daegu Gyeongbuk Institute of Science and Technology, Daegu, Republic of Korea; [5]Wellcome Trust Sanger Institute, Cambridge, United Kingdom; [6]Cancer Research United Kingdom Cambridge Institute, University of Cambridge, Cambridge, United Kingdom; [7]European Molecular Biology Laboratory - Monterotondo, Rome, Italy

**Abstract** The maternal-to-zygotic transition (MZT) marks the period when the embryonic genome is activated and acquires control of development. Maternally inherited factors play a key role in this critical developmental process, which occurs at the 2-cell stage in mice. We investigated the function of the maternally inherited factor Stella (encoded by *Dppa3*) using single-cell/embryo approaches. We show that loss of maternal Stella results in widespread transcriptional mis-regulation and a partial failure of MZT. Strikingly, activation of endogenous retroviruses (ERVs) is significantly impaired in Stella maternal/zygotic knockout embryos, which in turn leads to a failure to upregulate chimeric transcripts. Amongst ERVs, MuERV-L activation is particularly affected by the absence of Stella, and direct in vivo knockdown of MuERV-L impacts the developmental potential of the embryo. We propose that Stella is involved in ensuring activation of ERVs, which themselves play a potentially key role during early development, either directly or through influencing embryonic gene expression.

*For correspondence: jamie. hackett@embl.it (JAH); a.surani@ gurdon.cam.ac.uk (MAS)

†These authors contributed equally to this work

Competing interests: The authors declare that no competing interests exist.

## Introduction

Maternally inherited factors in the zygote play a critical role during early development (*Ancelin et al., 2016*; *Li et al., 2010*; *Wasson et al., 2016*). The maternal-to-zygotic transition (MZT) marks the time of transfer of developmental control to the embryo following activation of the zygotic genome (*Lee et al., 2014*; *Li et al., 2013*). In mice, the major wave of zygotic genome activation (ZGA) occurs at the late 2-cell stage (*Golbus et al., 1973*). The earliest zygotically transcribed genes are preferentially enriched in essential processes such as transcription, RNA metabolism and ribosome biogenesis (*Hamatani et al., 2004*; *Xue et al., 2013*). Other events that characterise early pre-implantation development include extensive erasure of global DNA methylation and dynamic changes in histone modifications (*Cantone and Fisher, 2013*).

Previous studies have shown that Stella, encoded by the *Dppa3* gene locus, is a maternally inherited factor that is required for normal pre-implantation development (*Bortvin et al., 2004*;

**eLife digest** When a sperm cell fertilizes an egg cell, this creates a single-celled embryo called a zygote that will go on to divide repeatedly throughout development. The zygote inherits the contents of the egg including many important proteins that initially control how the embryo develops. In mice, the embryo takes over control of development once the zygote has divided to form a two-celled embryo. This transition of control destroys the maternally inherited proteins and selectively activates zygotic genes and some DNA sequences called transposable elements that evolved from virus DNA.

One of the proteins inherited from the egg cell is called Stella. Embryos that lack Stella die within the first few cell divisions, which suggests that the protein is needed during the earliest stages of development. However, it is not clear what Stella's role in the early embryo is.

Huang, Kim et al. decided to investigate how Stella might affect the genes that are switched on during development by using a technique called RNA-seq to study egg cells and early embryos from mice. Two-celled embryos that lacked Stella could not activate a number of genes that produce proteins that are critical for development. They also failed to activate a group of transposable elements called endogenous retroviruses. In particular, a lack of Stella significantly reduced the activity of an endogenous retrovirus called MuERV-L.

Further experiments showed that MuERV-L is needed for normal embryonic development, and so suggests that transposable elements play important roles in this process. Future studies will aim to explore these roles in more detail. It will also be important to identify the genes that Stella targets in embryos, and to investigate the roles that similar maternally inherited proteins play in early embryonic development.

*Payer et al., 2003*). Stella is a small basic protein with a putative SAP-like domain and splicing-like factor; it also contains nuclear localisation and export signal and has the potential to bind to DNA and RNA in-vitro (*Nakamura et al., 2007*; *Payer et al., 2003*) (*Figure 1—figure supplement 1A*). Expression of Stella is high in oocytes, continues through pre-implantation development, and subsequently occurs specifically in primordial germ cells. Stella is also expressed in naïve embryonic stem cells, but is downregulated upon exit from pluripotency (*Hayashi et al., 2008*). Lack of maternal inheritance of Stella results in developmental defects; these manifest from the 2-cell stage onwards, and result in only a small proportion of embryos developing to the blastocyst stage. Notably, a maternally inherited pool of Stella in zygotic Stella knockout embryos, derived from Stella heterozygous females, allows development to progress normally. This indicates Stella has a key role during the earliest developmental events (*Payer et al., 2003*).

Stella was suggested to protect maternal pronuclei (PN) from TET3 mediated active DNA demethylation (*Nakamura et al., 2007*, *2012*) (*Figure 1—figure supplement 1B*). In the absence of Stella, zygotes display enrichment of 5hmC in both parental PN (*Wossidlo et al., 2011*), and an aberrant accumulation of γH2AX in maternal chromatin (*Nakatani et al., 2015*). In addition, Stella protects DNA methylation levels at selected imprinted genes and transposable elements (*Nakamura et al., 2007*). However, embryos depleted of maternal effect proteins known to regulate imprinted genes only exhibit developmental defects post-implantation (*Denomme and Mann, 2013*), implying that the 2-cell phenotype in Stella maternal/zygotic knockout (Stella M/Z KO) embryos is not primarily linked with imprint defects. Moreover, TET3 only partially contributes to DNA demethylation and its absence is compatible with embryonic development (*Amouroux et al., 2016*; *Peat et al., 2014*; *Shen et al., 2014*; *Tsukada et al., 2015*). Thus, what impairs pre-implantation embryonic development in the absence of maternal Stella remains unclear.

A significant number of transposable elements (TEs) are preferentially activated during early development and in a sub-population of mouse embryonic stem cells (*Gifford et al., 2013*; *Macfarlan et al., 2012*; *Rowe and Trono, 2011*). Notably, at the 2-cell (2C) stage in mouse development, there is selective upregulation of endogenous retrovirus (ERV), which are a subset of TEs characterised by the presence of LTRs that mediate expression and retrotransposition (*Kigami et al., 2003*; *Ribet et al., 2008*). Growing evidence suggests that activation of some TEs

has important biological functions during early development (*Beraldi et al., 2006*; *Kigami et al., 2003*). TEs can regulate tissue-specific gene expression or splicing through their exaptation as gene regulatory elements, and may also play a key role during speciation (*Böhne et al., 2008*; *Gifford et al., 2013*; *Rebollo et al., 2012*). TEs additionally drive expression of genes directly by acting as alternative promoters that generate chimeric transcripts, which include both TE and protein-coding sequences (*Macfarlan et al., 2012*; *Peaston et al., 2004*). Thus the expression of TEs may be functionally important during early embryo development and understanding the regulation of TEs themselves is therefore of great interest.

We adopted an unbiased approach to investigate the role of Stella during mouse maternal-to-zygotic transition, using single-cell/embryo RNA-seq analysis of mutant embryos (*Deng et al., 2014*; *Xue et al., 2013*; *Yan et al., 2013*). We find Stella M/Z KO 2-cell embryos fail to upregulate key zygotic genes involved in regulation of ribosome and RNA processing. Furthermore, the absence of Stella results in widespread misregulation of TEs and of chimeric transcripts that are derived from these TEs. In particular, Stella M/Z KO embryos exhibit a general failure to upregulate MuERV-L transcripts and protein at the 2-cell stage. Our perturbation data is consistent with MuERV-L playing a functionally important role during pre-implantation embryonic development, implying that MuERV-L is amongst the critical factors affected by Stella in early embryos.

## Results

### Stella M/Z KO embryos are transcriptionally distinct from wild type embryos

We used Stella knockout mice (*Payer et al., 2003*), to collect mutant oocytes and embryos for single-cell/embryo RNA-seq with modifications (*Tang et al., 2010*). Stella knockout (KO) oocytes and Stella maternal/zygotic knockout (Stella M/Z KO, KO) 1-cell and 2-cell embryos, lacking both maternal and zygotic Stella, were harvested. Strain matched wild type (WT) oocytes and embryos served as controls (*Figure 1A*; *Figure 1—figure supplement 2A*). Following normalisation, potentially confounding technical factors were controlled for using Removal of Unwanted Variation (RUV) (*Risso et al., 2014*) (*Figure 1—figure supplement 2B*, *Figure 1—source data 1*, *Supplementary file 1*).

Principal component (PC) analysis revealed that the first PC represents progression from oocyte to the 2-cell stage (*Figure 1B*). The largest separation is observed between oocyte/1-cell and 2-cell embryos, consistent with the degradation of maternal transcripts and activation of the zygotic genome at MZT. The second and third PCs capture the separation between WT and KO samples, suggesting distinct genome-wide gene expression changes between WT and KO embryos. Furthermore, although KO oocytes and 1-cell embryos exhibit some separation from their WT counterparts, the difference is more clearly pronounced at the 2-cell stage. Notably, Stella M/Z KO 1-cell and 2-cell embryos clustered separately from each other, suggesting that Stella M/Z KO defects at the 2-cell stage are not simply due to delayed developmental progression (*Figure 1B*). Differential gene expression analysis between WT and KO samples identified 5881 misregulated genes (adjusted p-value<0.05) across the developmental stages with 360 genes overlapping across all stages (*Figure 1C*, *Figure 1—source data 2*).

### Stella M/Z KO embryos are impaired in maternal-to-zygotic transition

Next, we compared the differentially expressed genes (DEG) against a public database of early mouse embryonic transcriptomes (DBTMEE) (*Park et al., 2015*). This dataset categorises genes by specific expression pattern at any developmental stage. Strikingly, we found significant enrichment of maternal transcripts (maternal RNA, minor zygotic genome activation and 1-cell transient genes) and depletion of zygotic transcripts (major zygotic genome activation, 2-cell transient and mid pre-implantation activation) in the Stella M/Z KO 2-cell embryos compared to WT (*Figure 1D*). While Stella KO oocytes are transcriptionally distinct from WT (*Figure 1B*), Stella KO oocytes do not exhibit overt abnormalities (*Bortvin et al., 2004*; *Payer et al., 2003*). Furthermore, the apparent enrichment in maternal transcripts only manifests at the 2-cell stage, suggesting the observed transcriptional differences are not inherited from aberrant transcripts in the Stella KO oocyte but are likely linked to failed downregulation of these maternal transcripts.

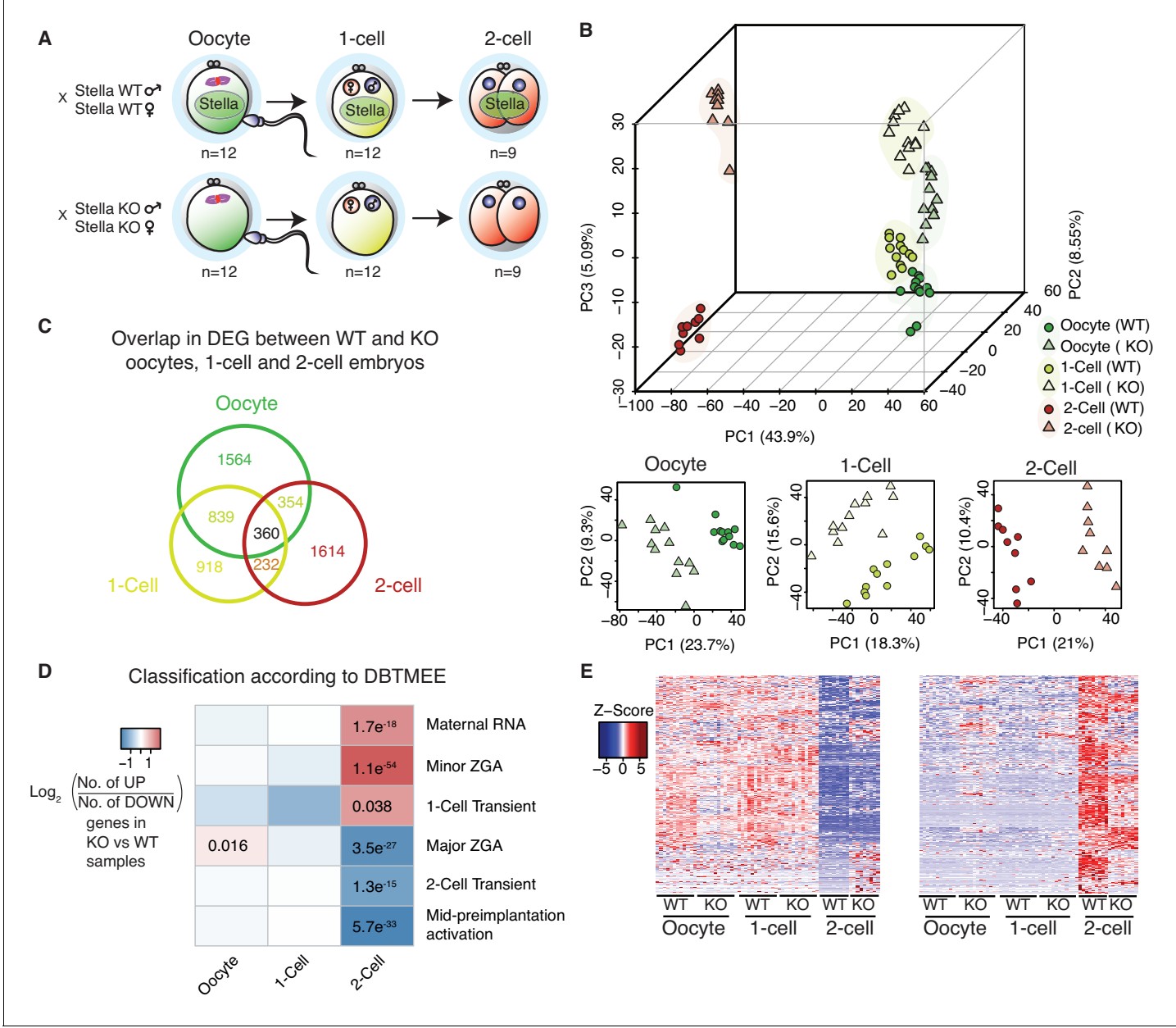

**Figure 1.** Stella M/Z KO embryos are impaired in maternal-to-zygotic transition. (**A**) A schematic illustration of the single-cell / embryo RNA-seq experimental setup. The total number of oocytes and embryos collected are indicated. The colour scheme represents the transition from maternal (green) to zygotic (red) transcripts. Maternal Stella is represented in green circle. (**B**) A score plot of the first three principal components for 66 cells using gene counts. The lower panels represent the score plots of the first two principal components using cells belonging to a specific developmental time point. The developmental time points are indicated by colour and the genotypes for Stella are indicated by shape. (**C**) A Venn diagram illustrating the overlap of differentially expressed genes (DEG) between WT and KO oocytes, 1-cell and 2-cell embryos (adjusted p-value<0.05) (*Figure 1—source data 2*). (**D**) The heatmap represents the $\log_2$ ratio of the number of upregulated to downregulated genes in KO compared to WT at oocytes, 1-cell and 2-cell stage, belonging to a given cluster of DBTMEE (*Park et al., 2015*). Fisher's exact test performed and statistically significant p-values are stated. (**E**) A time-series clustering of gene expression dynamics across oocyte to 2-cell embryo (see Materials and methods). (Left) Heatmap shows a cluster of WT maternal transcripts (ED), which are differentially expressed in KO samples (adjusted p-value<0.05, *Figure 1—source data 3*). (Right) Heatmap shows a cluster of WT zygotically activated genes (ZAG) (EU), which are differentially expressed in KO samples (adjusted p-value<0.05, *Figure 1—source data 4*). Also see *Figure 1—figure supplements 1–2* and *Figure 1—source data 1–4*.

The following source data and figure supplements are available for figure 1:

**Source data 1.** Gene counts for WT and KO oocyte, 1-cell and 2-cell embryos.

*Figure 1 continued*

**Source data 2.** List of differentially expressed genes between WT and KO samples at oocyte, 1-cell and 2-cell stage, from single cell/embryo RNA-seq analysis.
**Source data 3.** List of maternal transcripts, WT class = ED, which are differentially expressed in KO samples.
**Source data 4.** List of zygotically activated genes (ZAG), WT class = EU, which are differentially expressed in KO samples.
**Figure supplement 1.** Stella protein domains.
**Figure supplement 2.** Quality control of single-cell / embryo RNA-seq.

To independently characterise differences in gene expression dynamics between WT and KO embryos during MZT, genes were clustered based on their pattern of expression in WT oocytes, 1-cell and 2-cell embryos. We identified a cluster of maternal transcripts (ED), defined as highly expressed in WT oocytes and 1-cell and with reduced expression at the 2-cell stage, that are differentially expressed in KO samples (n = 849, adjusted p-value<0.05) (*Figure 1E*, *Figure 1—source data 3*). Of these genes, 37.7% did not show the expected reduction in expression in Stella M/Z KO 2-cell embryos. Furthermore, we detected a group of zygotically activated genes (ZAG) (EU) with low expression in WT oocytes and 1-cell embryos and increased expression in 2-cell embryos, which are differentially expressed in KO samples (n = 698, adjusted p-value<0.05). Moreover, 39.5% of this group of genes exhibited significantly dampened 2-cell activation in the absence of maternal and zygotic Stella (*Figure 1E*, *Figure 1—source data 4*). The combined analysis suggests that Stella M/Z KO embryos exhibit partial impairment in maternal-to-zygotic transition.

## Stella M/Z KO 2-cell embryos fail to upregulate essential ribosomal and RNA processing genes

Gene ontology analysis of genes more highly expressed in KO than WT 2-cell embryos reveals an enrichment of chromatin modifiers, and genes involved in microtubule based processes and response to DNA damage (*Figure 2A*, *Figure 2—source data 1*), consistent with a previous observation of abnormal γH2AX enrichment in maternal PN (*Nakatani et al., 2015*). In addition, downregulated genes in Stella M/Z KO 2-cell embryos are enriched for RNA processing and ribosome biogenesis, with an overall depletion of genes associated with ribosomes in the KEGG pathway and related gene ontology (*Figure 2B*, *Figure 2—figure supplement 1*). The findings from single-cell/ embryo RNA-sequencing were validated by independent qRT-PCR analyses. Consistently, genes associated with chromatin modifiers (*Rbbp7* and *Kdm1b*) and DNA damage (*Fam175a* and *Brip1*) were more highly expressed in KO embryos (*Figure 2C*), while those associated with ribosome biogenesis (*Nop16*, *Emg1*), RNA binding (*Larp1b*) and cell cycle regulator (*Cdc25c*) showed lower expression in KO 2-cell embryos (*Figure 2D*). To explore potential regulatory targets of Stella, a genome-wide co-expression network analysis was performed across the WT dataset. Overall, 910 genes showed correlated expression patterns with *Dppa3* (Pearson correlation >|0.9|) (*Figure 2— source data 2*). Of these, 710 genes are positively correlated while 200 genes are negatively correlated with *Dppa3*. Genes with a positive expression correlation with *Dppa3* are significantly enriched for RNA processing and RNA splicing (*Figure 2E*), including *Snrpd1* and *Snrpb2*, both core components of the spliceosome (*Figure 2F*).

Overall, our analysis suggests that the Stella M/Z KO 2-cell embryos do not effectively transition from maternal to zygotic control, as judged by the aberrant enrichment of maternal transcripts and depletion of zygotic transcripts. Genes regulating essential processes such as RNA processing are depleted in Stella M/Z KO 2-cell embryos and *Dppa3* exhibits positive expression correlation with genes enriched in these processes. Notably, most genes however do undergo appropriate MZT changes in mutant embryos, arguing against a general developmental delay. Taken together, this supports a hypothesis whereby Stella plays an important role in upregulating the expression of a subset of genes that are essential during zygotic genome activation.

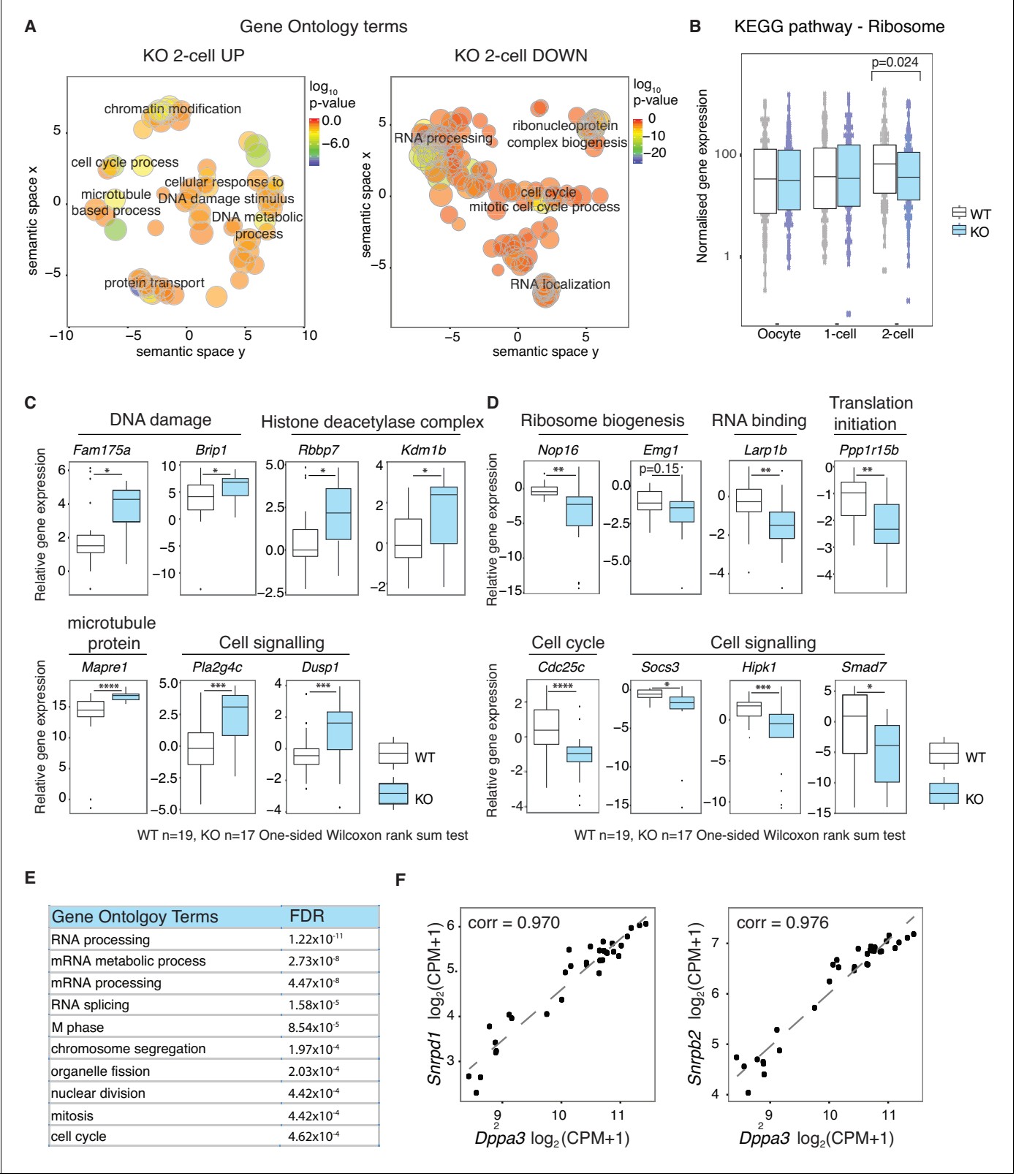

**Figure 2.** Stella is associated with the activation of essential zygotic genes. (A) Revigo plots (*Supek et al., 2011*) of a selection of gene ontology (GO) terms enriched in 2-cell DEG (*Figure 2—source data 1*). The colour of the circle represents the log₁₀ p-value. Semantic space clusters GO terms of similar functions together. (B) A boxplot of normalised counts of genes belong to the Ribosome KEGG pathway between WT and KO across the developmental stages. Two-sided Wilcoxon rank sum test performed between WT and KO and statistically significant p-value is stated. (C and D) shows
*Figure 2 continued on next page*

*Figure 2 continued*

boxplots of single-embryo qRT-PCR validation of RNA-seq in WT (white) and KO (light blue) 2-cell embryos. All genes are normalised to housekeeping genes, relative to one WT embryo and log$_2$ transformed. (C) shows genes significantly upregulated and (D) downregulated in KO relative to WT 2-cell embryos (p<0.05 = *; p<0.01 = **, p<0.001 = *** and p<0.0001 = ****). (E) A table of the top 10 GO terms enriched in 710 genes whose expression levels are positively correlated with *Dppa3* (Pearson's correlation coefficient >0.9), identified from the genome-wide co-expression network analysis (*Figure 2—source data 2*). (F) Scatter plots of the expression of *Snrpd1* and *Snrpb2* against *Dppa3*. Gene expression was log$_2$ transformed (counts per million + 1) and Pearson's correlation analysis was performed. Also see *Figure 2—figure supplement 1* and *Figure 2—source data 1–2*.

The following source data and figure supplement are available for figure 2:

**Source data 1.** List of gene ontology terms enriched in differentially expressed genes between WT and KO 2-cell embryos.
**Source data 2.** Genome-wide *Dppa3* co-expression network analysis.
**Figure supplement 1.** Ribosome associated genes are depleted in Stella M/Z KO 2-cell embryos.

## Expression of TEs are dysregulated in Stella KO oocytes and Stella M/Z KO embryos

Stella has been shown to affect the DNA methylation of particular TEs in zygotes (*Nakamura et al., 2007*) and primordial germ cells (*Nakashima et al., 2013*). Since many TEs are specifically activated during early development, we investigated their expression in WT and KO oocytes and embryos. We remapped the single-cell/embryo RNA sequencing data to obtain read counts for class I TEs (retrotransposons) (*Figure 3—source data 1–2*). To get an overview of the extent of TE activation, we calculated the fraction of transcripts mapped to TEs as a ratio of total mapped reads. We found a dramatic increase in the ratio of reads mapped to TEs, primarily of the LTR class, at the 2-cell stage compared to oocyte and 1-cell, suggesting widespread TE activation coincident with MZT (*Figure 3A*, *Figure 3B*). Further inspection revealed that activation of the LTR class is dominated by upregulation of ERVL and ERVL-MaLR families, while LINE and SINE classes are primarily accounted for by L1 and Alu families, respectively (*Figure 3—figure supplement 1A*). Moreover, principal component analysis showed developmental progression can be clearly captured by TE expression alone along PC1 (*Figure 3C*), similar to recent studies identifying stage-specific transcription initiation of ERV expression in early human embryos (*Göke et al., 2015*; *Grow et al., 2015*). In addition, WT and KO oocytes and embryos can be partially separated in individual PCAs along each of the developmental stages, signifying differences in expression of TEs between WT and KO samples.

## Stella is associated with the expression of specific TE

We identified 'maternal TEs', defined as TEs expressed higher in WT oocytes than 2-cell embryos and 'zygotic TEs', as TEs expressed higher in WT 2-cell embryos than oocytes. To characterise the disparities between TE expression in WT and KO, we compared differentially expressed TEs at the 2-cell stage, with maternal and zygotic TEs (*Figure 3D*). Strikingly, TEs upregulated in Stella M/Z KO 2-cells are significantly enriched for maternal TEs while TEs downregulated in Stella M/Z KO 2-cells are associated with zygotic TEs. Furthermore, Stella M/Z KO 2-cell embryos display lower activation of LTR, specifically the endogenous retrovirus (LTR-ERVL) family, with small relative enrichment for LINE and SINE classes (*Figure 3E*, *Figure 3—figure supplement 1B*).

A particular element of interest is MuERV-L (*Kigami et al., 2003*; *Ribet et al., 2008*), which encodes a canonical retroviral *Gag* and *Pol*, flanked by 5' and 3' LTR (also known as MT2_Mm). We observed a selective upregulation of MuERV-L-Int and MT2_Mm transcripts at the 2-cell stage, but this was significantly reduced in Stella M/Z KO embryos (*Figure 3F*). Many MuERV-L elements have an open reading frame and we therefore tested protein levels using a MuERV-L Gag antibody. This antibody was validated by showing staining overlaps well with a MuERV-L reporter ESC line – 2C::tdTomato (*Macfarlan et al., 2012*), and is specific to 2-cell embryos whilst not detected in metaphase II oocytes (*Figure 3—figure supplement 1C*). Strikingly, immunofluorescence (IF) staining showed a significantly lower signal and loss of perinuclear localisation of MuERV-L Gag (*Ribet et al., 2008*) in Stella M/Z KO 2-cell embryos compared to WT, implying Stella influences both transcript and protein levels of MuERV-L (*Figure 3G*).

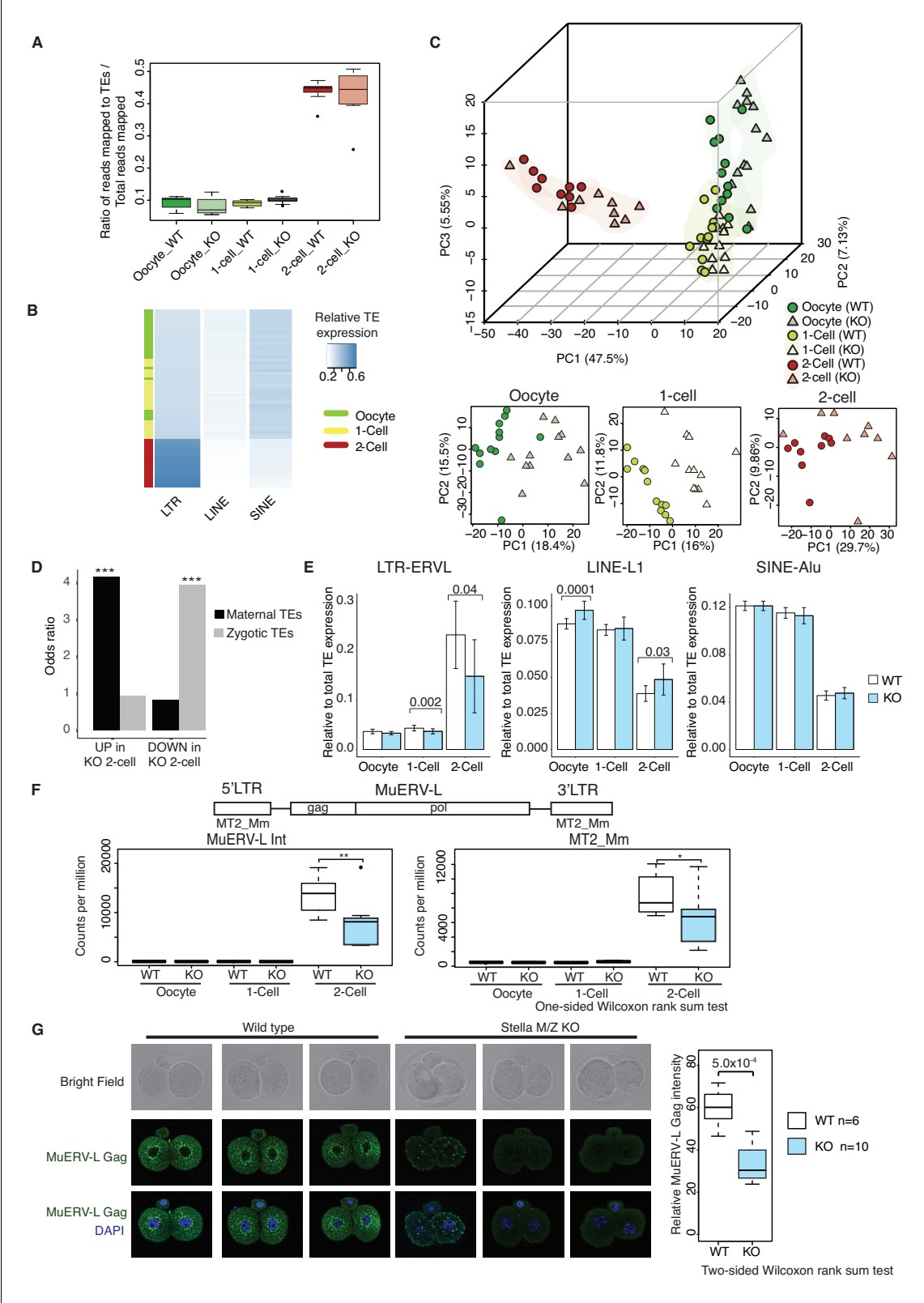

**Figure 3.** TEs are mis-expressed in the absence of Stella. (**A**) Top panel shows boxplots of the ratio of reads mapped to TEs to total reads mapped in WT and KO oocytes, 1-cell and 2-cell embryos. The number of reads mapped to TEs are based on uniquely and multi-mapped counts (*Figure 3— source data 2*). (**B**) A heatmap of the relative expression of LTR, LINE and SINE in oocyte (green), 1-cell (yellow) and 2-cell embryos (red). Blue indicates higher expression and white indicates lower expression. Samples are clustered by row. (**C**) A score plot of the first three principal components for 66

*Figure 3 continued on next page*

*Figure 3 continued*

cells using uniquely mapped TE counts (*Figure 3—source data 1*). The lower panels represent the score plots of the first two principal components using cells belonging to a specific developmental time point. The developmental time points are indicated by colour and the genotypes of Stella are indicated by shape. (**D**) A bar chart of the odds ratio of TEs up and downregulated in KO 2-cell embryos compared to WT, intersected with 'maternal TEs' and 'zygotic TEs'. For maternal TEs enriched in TEs upregulated in KO 2-cell, ***p=$5.35 \times 10^{-6}$; for zygotic TEs enriched in TEs downregulated in KO 2-cell, ***p=$8.13 \times 10^{-7}$. (**E**) Bar plots showing the relative expression of TE families (LTR-ERVL, LINE1-L1 and SINE-Alu) in WT (white) and KO (light blue) oocytes, 1-cell and 2-cell embryos, data analysed from single-cell / embryo RNA sequencing. Two-sided Wilcoxon rank sum test performed between WT and KO samples and statistically significant p-values stated. (**F**) Top is an illustration of the structure of the full-length MuERV-L element flanked by 5' and 3' LTRs. Bottom is boxplots of the relative expression (counts per million) of MuERV-L Int and MT2_Mm transcript in WT (white) and KO (light-blue) oocytes, 1-cell and 2-cell embryos ($p<0.05$ corresponds to * and $p<0.01$ corresponds to **). (**G**) Immunofluorescence staining against MuERV-L Gag antibody in 2-cell embryos. (Left) The top panel shows bright-field, middle panel is MuERV-L Gag (green) and bottom panel are merged images of MuERV-L and DAPI, which counterstains DNA. Representative projections of z-stacks are shown for WT and KO embryos. (Right) A boxplot of the z-stack quantifications of the relative fluorescence intensity of MuERV-L Gag protein between WT and KO 2-cell embryos. Also see *Figure 3— figure supplements 1–3* and *Figure 3—source data 1–2*.

The following source data and figure supplements are available for figure 3:

**Source data 1.** Uniquely mapped transposable element (TE) counts.

**Source data 2.** Uniquely and multi-mapped transposable element (TE) counts.

**Figure supplement 1.** TEs are mis-expressed in the absence of Stella.

**Figure supplement 2.** *Dppa3* does not affect MuERV-L expression in mESCs.

**Figure supplement 3.** Quality control of TE reads mapping and normalization.

To assess whether Stella directly binds to specific transcripts / TEs we turned to ESC since obtaining sufficient 2-cell embryos for this analysis is intractable. Chromatin immunoprecipitation-seq (ChIP-seq) of HA-tagged Stella resulted in minimal peak calls (n = 56) (*Supplementary file 2*), implying Stella does not efficiently bind DNA (including MuERV-L) in ESC. Consistently, neither overexpression nor knockdown of *Dppa3* in mESC had a marked effect on the expression of MuERV-L (*Figure 3—figure supplement 2*). These results suggest that Stella-mediated control of MuERV-L is likely specific to the unique chromatin/cellular context of early embryos. Notably Stella has the potential to bind both RNA and DNA and could thus affect TEs through multiple modes. Overall, our data suggests Stella plays a key role in regulating a subset of TEs specifically in 2-cell embryos, which includes promoting strong activation of MuERV-L elements.

## Expression of a subset of TEs is positively correlated with their nearest gene

Next we explored the relationship between TE and protein-coding gene expression in WT and KO samples. First, we considered the intersection between differentially expressed genes and genes that are neighbours of misregulated TEs at the 2-cell stage, and discovered a highly significant overlap of 12% between the two populations (p=$3.19 \times 10^{-87}$) (*Figure 4A*). Notably, a greater number of TEs downregulated in KO 2-cell embryos are located within 10 kb of the transcriptional start site of zygotically activated genes (ZAG, n = 698) than expected by chance ($p<10^{-4}$) (*Figure 4B*). Altogether, this suggests perturbations in TE expression may be linked to neighbouring mRNA expression levels in Stella M/Z KO embryos. To determine if this relationship is applicable genome-wide, we performed global expression correlation analysis between a TE and its nearest gene. To eliminate confounding factors, we excluded TEs that overlap directly with an exon. Notably, we detected 387 TEs whose expression levels were positively correlated with their nearest gene (Spearman's correlation $>0.7$) while 224 TEs were negatively correlated (Spearman's correlation $<-0.7$) (*Figure 4—figure supplement 1A*). This relationship was validated using single-embryo qRT-PCR at four genomic loci, which demonstrated a significant correlation between the expression of a TE and its downstream gene (*Figure 4C*). These loci were selected to represent a range of LTR class elements; RMER19A, ORR1A2-Int and MT2B are further categorised into ERVK, ERVL-MaLR and ERVL families

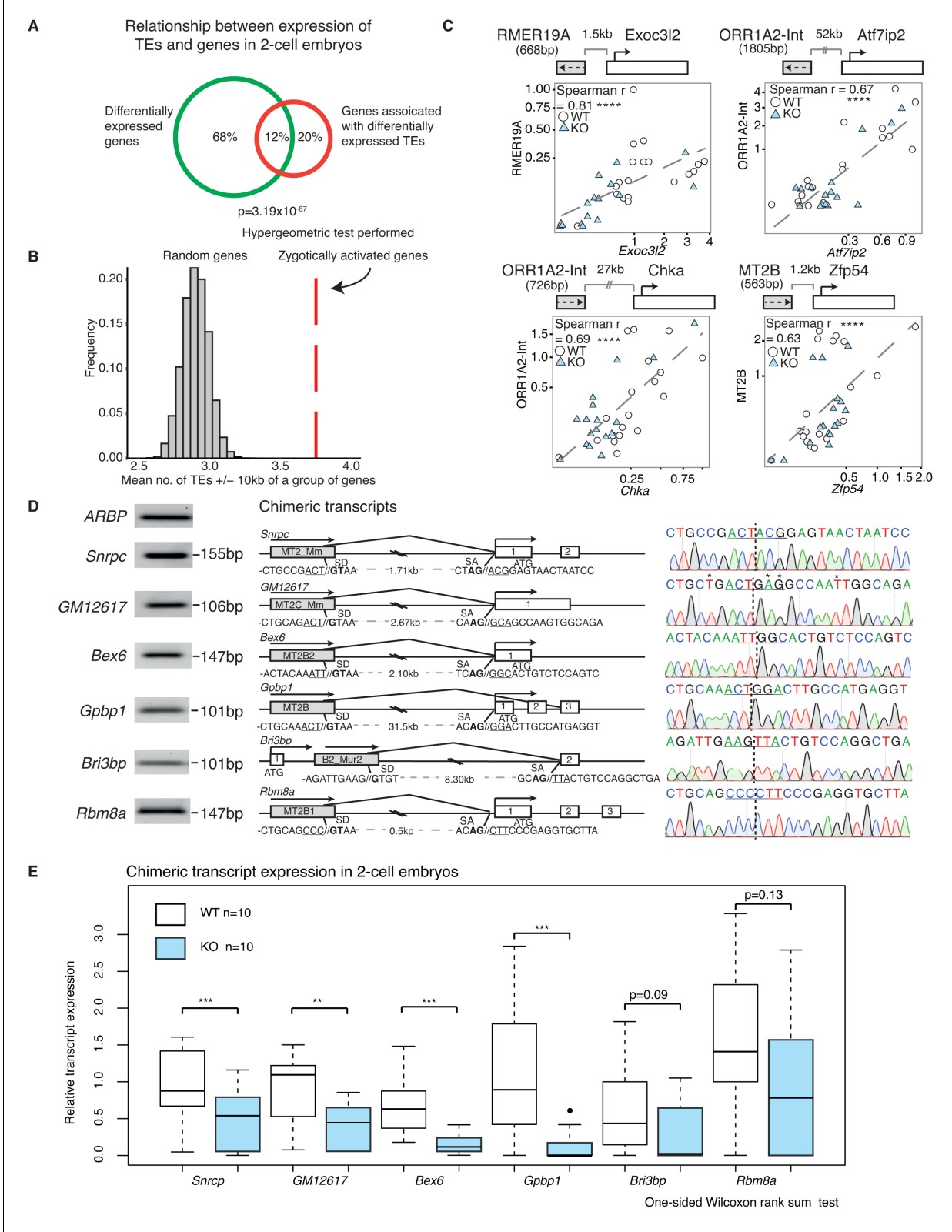

**Figure 4.** A positive correlation between the expression of a subset of TEs and their nearest gene. (**A**) A Venn diagram illustrating a significant overlap between differentially expressed genes and genes within ±20 kb of differentially expressed TEs in WT v KO 2-cell embryos. (**B**) A histogram of the mean number of depleted TEs in Stella M/Z KO 2-cell embryos within ±10 kb of the TSS of a group of genes. The red line represents zygotically activated genes (ZAG, n = 698) that belong to WT class EU and differentially expressed in KO samples (adjusted p-value<0.05) (**Figure 1—source data**

*Figure 4 continued on next page*

*Figure 4 continued*

4). The grey bars represent each of 10,000 random sets of 698 genes. (**C**) Top is schematic illustrations of the relationship between the TE and downstream gene, where grey box represents the TE and its orientation; white box represents the gene and its orientation. Bottom is scatter plots of qRT-PCR expression in 2-cell embryos. WT (white circle) n = 19 and KO (light blue triangle) n = 17. Spearman's correlation analysis was performed and p<0.0001 corresponds to ****. (**D**) Characterisation of chimeric transcripts. (Middle) A schematic illustration of the alternative splicing event that occurred between the TE and the downstream gene. SD = splice donor site and SA = splice acceptor site. The endogenous starting codon (ATG) is indicated. (Left) The PCR product of the chimeric transcript where the forward primer originates from the TE and reverse primer originates from the gene. The PCR product is significantly shorter than the annotated distances between them, confirming a splicing event. (Right) Sanger sequencing chromatographs from the PCR product for each chimeric transcript, confirming the site of chimeric junction. * Indicates a different nucleotide to annotation. (**E**) Box plots of single-embryo qRT-PCR of the relative expression of chimeric transcripts in WT (white) and KO (light blue) 2-cell embryos. p<0.01 corresponds to ** and p<0.001 corresponds to ***. All expression data are normalised to three housekeeping genes and relative to 1 WT embryo. Also see *Figure 4—figure supplement 1*.

The following figure supplement is available for figure 4:

**Figure supplement 1.** A subset of TEs is positively correlated with its nearest genes.

respectively. Interestingly, 2 of the genes, *Chka* and *Zfp54*, have been implicated in regulation of early embryonic development (*Albertsen et al., 2010*; *Wu et al., 2008*). Thus, the expression levels of a subset of TEs are highly correlated with the expression of their nearest gene.

Whilst association does not equate to causality, there are a number of ways TEs have been implicated in the regulation of gene expression (*Elbarbary et al., 2016*; *Thompson et al., 2016*). One possibility is the establishment of alternative splicing events, which create a junction between an expressed TE and a downstream gene exon, thus leading to the formation of a chimeric transcript (*Peaston et al., 2004*). Indeed, MuERV-L has been shown to form such fusion transcripts with 307 genes at the 2-cell stage (*Macfarlan et al., 2012*). Here, we find Stella M/Z KO embryos with the lowest MuERV-L expression also express lower levels of genes that make a chimeric junction with MuERV-L elements (*Figure 4—figure supplement 1B*). In contrast, the KO embryo with the highest MuERV-L expression expresses higher levels of these genes. Importantly, we next identified and validated a number of chimeric transcripts that are expressed in 2-cell embryos by PCR and sequenced across the chimeric junction (*Figure 4D*). Single-embryo qRT-PCR shows most of these chimeric transcripts are expressed at lower levels in KO compared to WT 2-cell embryos (*Figure 4E*). These chimeric transcripts include *Snrpc*, which encodes for a component of the ribonucleoprotein required for the formation of the spliceosome, and the RNA binding protein, *Rmb8a*. Our data implies a link between activation of some TEs and the emergent gene expression network, which is perturbed in Stella M/Z KO embryos due to altered TE expression. This is, in part, mediated by TEs directly regulating gene expression through chimeric transcripts.

## MuERV-L plays a functional role during pre-implantation development

We propose that the dysregulation in TE expression might directly contribute to the abnormal pre-implantation phenotype observed in Stella M/Z KO embryos. We focused on assessing the function of MuERV-L elements, as they represent one of the most highly activated TEs at the 2-cell stage (*Kigami et al., 2003*), as well as being amongst the most significantly downregulated in Stella M/Z KO embryos (*Figure 3F,G*). We considered that attenuated MuERV-L activation in Stella M/Z KO embryos could in turn disrupt activation of associated 2-cell transcripts driven by their LTRs (*Figure 4—figure supplement 1B*). Alternatively, MuERV-L mRNA/protein itself may also have a functional role during early embryonic development.

We detected 583 copies of the full-length MuERV-L element, containing the complete *gag* and *pol* genes, in the genome. This makes it experimentally challenging to completely eliminate MuERV-L activation in early embryos. Nevertheless, we utilised siRNA to knockdown the expression of MuERV-L elements in 1-cell embryos and monitored the effects on embryonic development (*Figure 5A*). We designed siRNA to target the consensus sequence of MuERV-L from the Dfam database (*Hubley et al., 2016*) (*Figure 5B*). Computational analysis reveals MuERV-L siRNA targets 80.5% of full-length MuERV-L elements with perfect match and 99.5% with ≤2 bp mismatches, while scramble siRNA has no targets with ≤2 bp mismatches. At 20 µM siRNA, we observed a slight

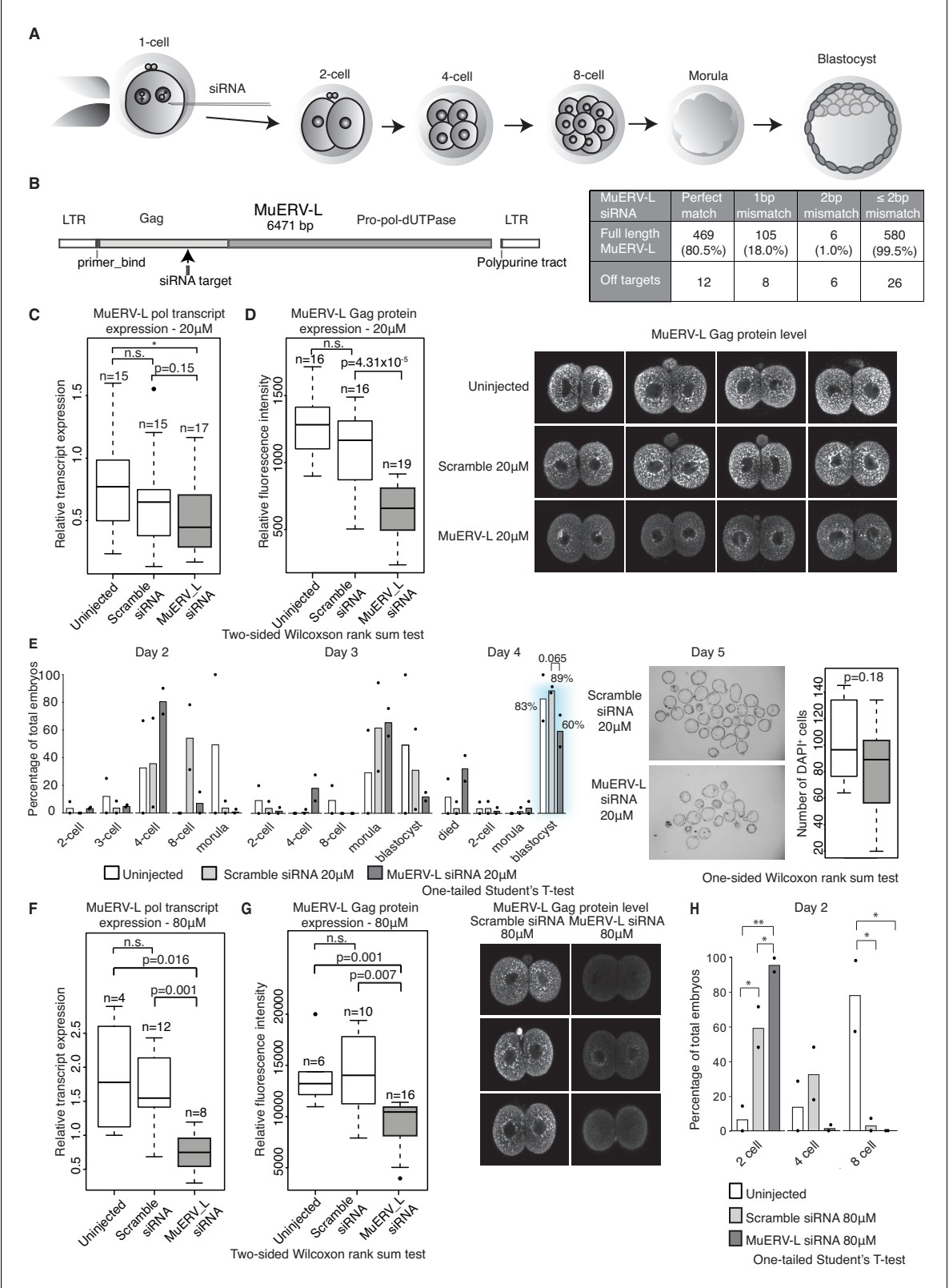

**Figure 5.** MuERV-L plays a functional role during pre-implantation development. (**A**) A schematic diagram illustrating the experimental setup. (**B**) Left is an illustration of the structure of the MuERV-L element (GenBank accession number: Y12713), depicting the exact site of siRNA target. Right is a table showing the number of full-length MuERV-L elements targeted by MuERV-L siRNA and the number of off-targets. (**C**) Box plots of single-embryo qRT-PCR of MuERV-L pol transcript and D) MuERV-L Gag protein (fluorescence) intensity in uninjected controls, 20 μM scramble siRNA or 20 μM MuERV-L

*Figure 5 continued on next page*

*Figure 5 continued*

siRNA injection 2-cell embryos. (Right) Representative z-stack projections of immunofluorescence (IF) staining against MuERV-L Gag in 2-cell embryos. Two-tail Wilcoxon Rank Sum Test was performed for both (**C**) and (**D**) and p<0.01 corresponds to *, not significant (n.s.) or are otherwise stated. (**E**) The effect of MuERV-L siRNA on developmental progression of pre-implantation embryos. (Left) Bar graphs illustrating the number of embryos observed at different stages on day 2–4 of in-vitro culture. Blue background indicates area of interest. Experiments were repeated twice and the circle dots represent the data points for each replicate. A total of n = 17, n = 58 and n = 53 embryos were analysed for uninjected, scramble and MuERV-L siRNA injections respectively. One-tailed Student's T-test performed and there were no statistical difference between uninjected, scramble or MuERV-L siRNA injected groups. (Middle) Day five bright field images of late blastocysts injected with scramble or MuERV-L siRNA. (Right) A box plot of IF quantification of number of DAPI$^+$ cells in the late blastocyst stage on day five injected with 20 µM of scramble (white) or MuERV-L (grey) siRNA. (**F** and **G**) are box plots of single-embryo qRT-PCR of MuERV-L pol transcript and MuERV-L Gag protein (fluorescence) intensity in uninjected control, 80 µM scramble siRNA and 80 µM MuERV-L siRNA injected 2-cell embryos. (Right) Representative z-stack projections of IF staining against MuERV-L Gag antibody in 2-cell embryos. Two-tail Wilcoxon Rank Sum Test was performed for both (**F**) and (**G**) and p<0.01 corresponds to *, not significant (n.s.) or are otherwise stated. (**H**) A bar graph plotting the number of embryos observed at each stage in day two in-vitro culture in uninjected embryos, 80 µM scramble (white) or MuERV-L (grey) siRNA injection. Experiments were repeated twice and the circle dots represent the data points for each replicate. A total of n = 17, n = 54 and n = 50 embryos were analysed for uninjected, scramble and MuERV-L siRNA respectively. One-tailed Student's T-test performed. p<0.05 corresponds to * and p<0.01 corresponds to **.

The following figure supplement is available for figure 5:

**Figure supplement 1.** Chimeric transcript expression in MuERV-L knockdown embryos.

decrease in MuERV-L transcripts (*Figure 5C*) but a significant reduction in MuERV-L Gag protein staining in 2-cell embryos (*Figure 5D*). These embryos exhibited a mild developmental delay at 4 to 8 cell stage progression on day 2, with fewer embryos reaching the blastocyst stage on day 4, and those that did were smaller and with fewer cells (*Figure 5E*). This suggests that a modest reduction in MuERV-L at the 2-cell stage has a notable effect on pre-implantation development.

There is nonetheless variability in MuERV-L expression at the 2-cell stage, as shown in other studies (*Ancelin et al., 2016*), and we therefore speculated that a higher concentration of siRNA might increase the repression efficiency of MuERV-L transcripts. Indeed, 80 µM MuERV-L siRNA was more effective (*Figure 5F and G*), and in turn resulted in >90% embryos arrested at the 2-cell stage (*Figure 5H*), albeit some indirect effects at higher concentrations of siRNA may contribute to the embryonic arrest. Notably, MuERV-L knockdown resulted in repression of some chimeric transcripts previously characterised in 2-cell embryos (*Figure 4D*, *Figure 5—figure supplement 1*). These data collectively suggest MuERV-L may have a functional role in embryonic development.

## Discussion

The maternal-to-zygotic transition is a complex process that entails extensive global transcriptional and epigenetic changes. Maternally inherited factors such as Stella play a role in this transition in mouse oocytes but the difficulties of obtaining sufficient materials has hindered investigations on how they contribute to MZT. Here, we use single-cell/embryo approaches and find that deletion of maternal and zygotic Stella affects MZT. Furthermore, we observed a failure to activate endogenous retrovirus (LTR-ERVL), and specifically MuERV-L elements. In parallel there is an accumulation of maternal transcripts and a failure to upregulate many zygotic transcripts. These events appear to be partially coupled, with altered LTR promoter activity leading to changes in gene expression, for example through the modulation of chimeric transcripts. In addition, we reveal the biological relevance of TE expression as knockdown of MuERV-L at the 2-cell stage hinders developmental progression, which likely contributes to the phenotype observed following loss of Stella.

Maternal mRNA degradation and zygotic genome activation are critical aspects during MZT (*Li et al., 2013*). Maternal mRNA clearance plays both a permissive and instructive role (*Tadros and Lipshitz, 2009*), while ZGA also promotes the final degradation of maternal transcripts. Stella has the potential to function as an RNA binding protein, which might facilitate the degradation of maternal transcripts (*Nakamura et al., 2007*; *Payer et al., 2003*). In addition, Stella KO oocytes exhibit transcriptional differences compared to WT oocytes. This is consistent with a previous study showing Stella KO oocytes are defective in transcriptional repression (*Liu et al., 2012*), which may be needed for dosage regulation of critical maternally inherited transcripts. Stella has also however been

demonstrated to be required beyond oogenesis (*Nakamura et al., 2007*). Maternally inherited factors likely influence the timing and specificity of gene expression during ZGA, which is essentially determined by interplay between the transcriptional machinery and histone modifications that influences chromatin accessibility (*Lee et al., 2014*).

The failure of zygotic transcript activation may be linked to the failure of zygotic TE activation. We detected extensive changes in global TE expression during MZT. This is accompanied by strong upregulation of the LTR-ERVL family in 2-cell embryos, which is consistent with a recent study showing that the chromatin surrounding ERVL is in a highly accessible state at the 2-cell stage (*Wu et al., 2016*). LTR-ERVL upregulation is significantly reduced in the absence of maternal Stella, suggesting that Stella may facilitate the activation of many ERVs. The expression changes are validated at the protein level for a specific LTR-ERVL element, MuERV-L. Importantly, the decrease in MuERV-L is unlikely to be a secondary effect of 2-cell embryonic arrest, as maternal *Kdm1a* (LSD1) mutant embryos, which result in pan 2-cell arrest maintain normal MuERV-L transcript expression (*Ancelin et al., 2016*). Notably, whilst many epigenetic mechanisms have been identified for the suppression of ERVs (*Gifford et al., 2013*), we have identified a factor associated with the activation of a subset of ERVs during mammalian MZT.

The question remains, how does Stella regulate TE expression? In primordial germ cells, Stella directly binds to LINE1 and IAP elements to facilitate DNA demethylation of these targets (*Nakashima et al., 2013*). Whilst experimentally challenging, it would be pertinent to determine potential DNA binding targets of Stella in 2-cell embryos. At the same time, function of Stella at the RNA level cannot be excluded, or it could act by altering the epigenetic status of cells at this time. For example, the H3K9 methyltransferase *Setdb1* is aberrantly enriched in Stella M/Z KO 2-cell embryos, and this has been shown to suppress ERVs and chimeric transcripts expression (*Eymery et al., 2016*; *Karimi et al., 2011*; *Kim et al., 2016*). Since a human orthologue of *Dppa3* is expressed in ESCs, germ cells and pre-implantation embryos (*Bowles et al., 2003*; *Deng et al., 2014*; *Payer et al., 2003*), it would be of interest to know whether *DPPA3* plays a similar role during human pre-implantation development; for example, on the stage specific activation endogenous retroviruses (*Göke et al., 2015*; *Grow et al., 2015*).

TEs have evidently been co-opted for the regulation of mammalian development as exemplified by the domestication of the retroviral *env* gene, which is essential for placental development (*Dupressoir et al., 2012*). Here, we have shown that the expression of a subset of TEs are intimately linked to its nearest gene during early embryonic development. As we have shown in several cases, TEs could regulate gene expression through chimeric transcripts, which are misregulated in the absence of Stella. The impact of transcriptional inhibition of ERV programme and associated chimeric transcripts during MZT, and their subsequent effects on development merits further investigation. Intriguingly, we discovered that reducing MuERV-L mRNA/ protein levels at the 2-cell stage affects development. This could suggest that the presence of MuERV-L Gag protein has a functional role during early mouse development, similar to a recent report illustrating a role of endogenous LTR viral-like particles in human blastocyst development (*Grow et al., 2015*). It is also possible that MuERV-L siRNA KD may downregulate a critical subset of chimeric transcripts that include MuERV-L sequence or, could feedback to target transcriptional repression to MuERV-L loci through small RNA pathways (*Castel and Martienssen, 2013*). Disentangling these possibilities warrants further investigation.

In conclusion, we have revealed that Stella is involved in orchestrating the transcriptional changes and activation of endogenous retroviral (ERV) elements during maternal-to-zygotic transition. The appropriate expression of LTR-ERV driven zygotic genes and specific ERV elements (MuERV-L) in turn contribute towards normal early embryonic development in mice.

## Materials and methods

### Experimental methods

#### Collection of mouse oocytes and embryos

All husbandry and experiments involving mice were carried out according to the Home Office guidelines and the local ethics committee. Here, we refer to Stella as the protein encoded by the *Dppa3* gene. Stella knockout (KO) mice were generated as previously described (*Payer et al., 2003*) (RRID:

MGI:2683730) in a pure 129/SvEv strain. For RNA-seq, ovulated oocytes were collected from Stella KO or wild type (WT) females crossed with vasectomised male at 9am, embryonic day (E) 0.5. Stella KO females were crossed with Stella KO males to produce Stella maternal and zygotic knockout (Stella M/Z KO, KO) embryos, while Stella WT females were crossed with Stella WT males to obtain WT embryos. 1-cell and 2-cell embryos were collected at 4pm on E0.5 and 9am-3pm on E1.5 respectively. All oocytes and embryos were morphologically assessed to ensure only viable samples were collected.

## Single embryo RNA sequencing and RT-qPCR

Oocytes were incubated with 0.3 mg/ml hyaluronidase (Sigma, St. Louis, MO) to remove the cumulus cells. Zona pellucida was removed from oocytes and embryos prior to lysis. RNA were extracted from single oocytes, 1-cell embryos or whole 2-cell embryos; and cDNA were amplified as described previously with modifications (*Tang et al., 2010*). Between 1:10,000–1:60,000 dilution of ERCC RNA Spike-In mix (ThermoFisher Scientific, Hemel Hempstead, UK) was added to lysis buffer to estimate efficiency of amplification (*Jiang et al., 2011*). 1:10 dilution of the cDNA was used for RT-qPCR. For relative expression, all genes / TE transcripts were normalised to three housekeeping genes (ARBP, PPIA and GAPDH). For primer sequences see *Supplementary file 3*. For RNA-seq, cDNA were fragmented to ~200 bp with Focused-ultrasonicator (Covaris, Woburn, MA) and adaptor ligated libraries were generated using NEBNext Ultra DNA library Prep Kit for illumina (New England Labs, Ipswich, MA). Single-end 50 bp sequencing was performed with HiSeq1500 (Illumina, San Diego, CA) to an average depth of 13.5 million reads per sample.

## 2-cell embryo immunofluorescence staining

2-cell embryos were fixed in 4% paraformaldehyde (PFA) in PBS for 15min at room temperature (RT). After permeabilisation with IF buffer (0.1% Trition, 1% BSA in PBS) for 30min at RT, embryos were incubated with primary antibody overnight at 4°C. After washing 3x with IF buffer, embryos were incubated with corresponding secondary antibody and 1 μg/ml of DAPI (4',6'-diamidino-2-phenylindole) for DNA visualisation for 1 hr at RT. After washing, embryos were mounted in VECTA-SHIELD Mounting medium (Vector Laboratories, Peterborough, UK). For each experiment, WT and KO embryos were stained at the same time and processed identically using the same setting for confocal acquisition to allow comparison of relative fluorescence intensity. The following antibodies were used: rabbit anti-MuERV-L Gag (ER50102, Huangzhou HuaAn Biotechnology Co., LTD, China) (RRID:AB_2636876), 1:200; Alexa488 donkey anti-rabbit IgG (ab150073, Abcam, Cambridge,UK) (RRID:AB_2636877), 1:500.

## siRNA injections

B6CBAF1/J (F1) female mice were superovulated by injection of 5 International Units (IU) of pregnant mare's serum gonadotrophin (PMSG) (Sigma, St. Louis, MO), followed by 5 IU of human chorionic gonadotrophin (hCG) (Sigma, St. Louis, MO) after 48 hr and then mated with F1 male mice. Zygotes were harvested from oviducts 17–22 hr post hCG injection. Cumulus cells were removed by incubation with 0.3 mg/ml hyaluronidase in M2 medium (Sigma, St. Louis, MO). 20 μM or 80 μM of Stealth RNAi siRNA (ThermoFisher Scientific, Waltham, MA) were micro-injected into the cytoplasm of zygotes using a Femtoject 4i device (Ependorf, UK). The following are the siRNA sequences: scramble (sense: UUCCUCUCCACGCGCAGUACAUUUA) and MuERV-L (sense: GAAGAUAUGCCUUUCACCAGCUCUA). Injected embryos were cultured in M16 medium (Sigma, St. Louis, MO) in BD Falcon Organ Culture Dish at 5% $CO_2$ and 37°C for a total of 5 days. The number of surviving 2-cell embryos 24 hr after micro-injection represents the total number of embryo analysed per experimental group. 20 μM or 80 μM siRNA injections were repeated twice independently.

## Stella-HA chromatin immunoprecipitation

We tagged three copies of hemagglutinin (HA, 27 bp) to the C-terminus of Stella and used the high affinity HA antibody to pull down Stella bound chromatin. The Stella+HA construct was expressed in the absence of endogenous Stella by using Stella M/Z KO mESCs, with NANOG-HA and eGFP-HA as positive and negative controls respectively. The ChIP protocol was modified from a previously described Stella ChIP protocol (*Nakamura et al., 2012*). Briefly, cells were cross-linked with 1%

paraformaldehyde (PFA) for 8 min at RT, quenched with 200 mM glycine for 5 min and washed twice with PBS. Samples were suspended in 2 ml/IP of radio immunoprecipitation assay (RIPA) buffer (50 mM Tris-HCl, pH8, 150 mM NaCl, 1 mM EDTA, 1% NP40, 0.5% deoxycholate and 0.1% SDS) with 1:20 dilution of proteinase inhibitor cocktail (Roche, Welwyn Garden City, UK) and sonicated to an average length of 200–1000 bp. Dynabeads Protein G (ThermoFisher Scientific, Hemel Hempstead, UK) were pre-incubated with anti-mouse HA antibody (MMS-101P-500, Covance Research Products Inc, Denver, CA) (RRID:AB_291261) at 7 µg/IP for 1 hr at 4°C. The antibody-beads complex was then incubated with chromatin solution overnight at 4°C. Beads were washed twice with RIPA buffer, twice with high-salt wash buffer (20 mM Tris-HCl, pH8; 500 mM NaCl, 1 mM EDTA, 1% NP-40, 0.5% deoxycholate and 0.1% SDS), twice with LiCl wash buffer (250 mM LiCl, 20 mM Tris-HCl, pH8, 1 mM EDTA, 1% NP-40 and 0.5% deoxycholate) for 10 min at 4°C and a final wash with Tris-EDTA. Chromatin were eluted by adding 100 µl of SDS-Elution buffer (50 mM Tris-HCl (pH 7.5), 10 mM EDTA, 1% SDS) and incubated at 68°C for 10 min. Reverse-crosslink was performed with 100 µl of proteinase K (1 mg/ml) for 2 hr at 42°C and 5 hr at 68°C. Finally, DNA were extracted with phenol purification and precipitated with NaCl into pellets. DNA pellets were re-suspended in 11 µl of $H_2O$. We detected enrichments of NANOG in known binding regions using ChIP-qPCR, but not for eGFP, supporting the efficacy of the ChIP (data not shown). Stella-HA ChIP sequencing libraries were prepared with NEBNext Ultra Library Prep Kit for Illumina (New England Labs, Ipswich, MA) with ~8 ng DNA input. Single-end 50 bp sequencing was performed with HiSeq2500 (Illumina, San Diego, CA). Stella -HA ChIP seq enriched peaks are shown in *Supplementary file 2*.

## 2C::tdTomato ESC culture and *Dppa3* transfection

2C::tdTomato reporter (Addgene plasmid #40281) ESCs (*Macfarlan et al., 2012*) were cultured in GMEM (Life Technologies, Carlsbad, CA), 10% KnockOut Serum Replacement (ThermoFisher Scientific, Waltham, MA), 2 mM L-glutamine (Life Technologies), 0.1 mM MEM non-essential amino acids (Life Technologies), 100 U/ml penicillin and 100 µg/ml streptomycin (Life Technologies), 1 mM sodium pyruvate (Sigma), 0.05 mM 2-mercaptoethanol (Life technologies), LIF (1000 U/ml; ESGRO; Merck Milipore) with 2i (PD0325901 (1 µM; Stemgent, Cambridge, MA) and CHIR99021 (3 µM; Stemgent, Cambridge, MA). For *Dppa3* over-expression, a Tet-ON3G inducible vector containing *Dppa3* CDS was transfected into ESCs with Lipfectamine 2000 (ThermoFisher Scientific, Waltham, MA). 500 ng/ml of DOX was added on day 0. For *Dppa3* knockdown experiment, ON-TARGETplus SMARTpool (Dharmacon, Lafayette, CO) which contain four siRNA targeting *Dppa3* was transfected into ESCs using Lipofectamine RNAiMAX transfection reagent (ThermoFisher Scientific, Waltham, MA). The *Dppa3* siRNA SMARTpool includes the following siRNA sequences: GGAUGAUACAGACG UCCUA; UAGAUAGGAUGCACAACGA; AGAAAGUCGACCCAAUGAA; GAGUAUGUACGUUCUAA UU. ON-TARGETplus Non-targeting (scramble) siRNA (Dharmacon) was transfected as negative control. Td-tomato expression was analysed with BD LSRFORTESSA (BD Biosciences, San Jose, CA).

## Data processing and analysis

### Mapping reads for gene-level counts and data processing

Single-end reads were mapped to the *Mus musculus* genome (GRCm38) using GSNAP (version 2014-07-21) with default options (*Wu and Nacu, 2010*). To detect splice junctions in reads, we extracted known splice sites from the GTF file of GRCm38 provided by Ensembl (release 79). Uniquely mapped reads were counted for each gene using htseq-count (*Anders et al., 2015*). We assessed quality of cells following the criteria previously described (*Kolodziejczyk et al., 2015*) (*Figure 1—figure supplement 2A*). To remove unwanted variation between batches, we applied a generalised linear model to the gene-level counts using the RUVs function of the RUVSeq package of R with k = 4 (*Risso et al., 2014*) (*Figure 1—source data 1*). To perform principal component analysis, the batch-adjusted gene counts were normalised by the size factor estimated by the DESeq2 package of R and lowly expressed genes whose mean normalised counts are below 100 were filtered out. The normalised counts were log-transformed and a pseudo count of 1 was added. The prcomp function of R was applied to the log-transformed gene counts by enabling the scaling option.

## Differential expression analysis between WT and KO

From the batch-adjusted counts, we identified differentially expressed genes between WT and KO for each developmental stage using the DESeq function of the DESeq2 package of R with default options (*Love et al., 2014*). Genes that have an adjusted P-value less than a given FDR cutoff of 0.05 were considered as differentially expressed (*Figure 1—source data 2*).

## Clustering of time-series data

To cluster the dynamic expression profile of genes in WT, we defined nine classes according to any differences between adjacent time points: EE, ED, EU, DE, DD, DU, UE, UD, UU, where 'E' denotes equally expressed, 'D' denotes downregulated or 'U' denotes upregulated at a later time point. A gene is considered to show significant differences between adjacent time points ('D' or 'U') if the adjusted P-value estimated from DESeq function of the DESeq2 package with a default option is less than 0.1. Otherwise, the difference is called as 'E'. For example, 'ED' means (1) equally expressed between oocyte and 1-cell and (2) downregulated at 2-cell compared to 1-cell. Differential expression analysis was performed between WT and KO samples, adjusted p-value<0.05 is considered significantly different. WT class ED genes that are differentially expressed in KO samples are represented in the left heatmap (*Figure 1E*, *Figure 1—source data 3*). WT class EU genes that are differentially expressed in KO samples are represented in the right heatmap (*Figure 1E*, *Figure 1—source data 4*).

## Gene ontology analysis

Gene ontology (GO) analyses were performed using DAVID 6.7 (https://david.ncifcrf.gov/home.jsp) for differentially expressed upregulated or downregulated genes in KO compared to WT 2-cell embryos. Adjusted p-values from BP_Fat were plotted with ReviGO (*Supek et al., 2011*) allowing for medium similarity, with a selection of GO terms displayed (*Figure 2A*, *Figure 2—source data 1*).

## Genome-wide *Dppa3* gene co-expression analysis

A *Dppa3* co-expression network (*Bassel et al., 2011*) was constructed for genes expressed in WT oocyte, 1-cell, and 2-cell stages. The co-expression network can be represented as an adjacency matrix, $C$, with an edge being drawn between two genes $i$ and $j$ if the magnitude of the Pearson correlation coefficient between the $\log_2$ transformed CPM ($\log_2 (CPM + 1)$) was greater than a particular threshold value:

$$C_{i,j} = \begin{cases} \rho_{i,j}, & |\rho_{i,j}| \geq x, \\ 0, & |\rho_{i,j}| < x. \end{cases}$$

## Mapping reads and normalisation for TE counts

To investigate the expression of class I TE (retrotransposons), we remapped the sequencing data to the same genome by randomly keeping only one genomic alignment of multi-mapped reads using GSNAP. From the GTF file of RepeatMasker provided by UCSC table browser (downloaded at 2015-08-25), uniquely mapped reads were counted for each genomic region of the TE (annotated as repName in RepeatMasker) using htseq-count (*Figure 3—source data 1*). We also generated multi-mapped counts by considering both uniquely and multi-mapped reads. To this end, the optional NH tags of all alignments in the SAM files were set to one and mapped reads were counted for each TE using htseq-count (*Figure 3—source data 2*). We show multi-mapped reads can be unambiguously assigned to a class (repClass in RepeatMasker) and family (repFamily in RepeatMasker) 98% of the time, however, only 54% of the time at the level of an element (repName in RepeatMasker) (*Figure 3—figure supplement 3A*). For this reason, all analysis at the level of retrotransposon class and family are based on multi-mapped counts, while analysis at the level of individual transposable elements are based only on uniquely mapped counts. To adjust for different sequencing depths, we normalised the TE counts by the total number of reads mapped to TEs and used the within-sample normalised values as our estimates of the expression levels of TEs (*Figure 3B*). The dynamic expression profiles of TEs during MZT were unchanged even when we normalised the TE counts by the size factor estimated by the DESeq2 package of R (between-sample normalisation method) (*Figure 3—figure supplement 3B*).

## Stella ChIP-Seq analysis

We mapped single-end reads of ChIP-seq experiments to the *Mus musculus* genome (mm10) using Bowtie2 (version 2.2.7) with default options (–local) (*Langmead and Salzberg, 2012*). After including only uniquely mapped reads, peak calling was performed by HOMER (*Heinz et al., 2010*) (version 4.8.1) with default options (findPeaks –style factor), where the input sequencing run was used as a control. The peaks were associated with nearby genes (mm10) by using HOMER annotatePeaks.pl.

## Correlation analysis between genes and TEs

Uniquely mapped BAM files were uploaded into SeqMonk (Version 0.32.0) (https://www.bioinformatics.babraham.ac.uk/projects/seqmonk/). The edgeR package (*McCarthy et al., 2012*) was used to identify differently expressed genes and TEs between WT and KO 2-cell embryos. We performed an intersection between differentially expressed genes and genes ±20 kb of differentially expressed TEs. A hypergeometric test was used to determine the significance of the overlap. For genome-wide analysis, we calculated Spearman's rank correlation coefficient between a TE and its nearest neighbouring gene using the expression profile of 66 cells. We excluded all TEs overlapped with any exons including UTRs, allowing the max gap of 51 bp equal to the read length. The maximum distance between genes and TEs was set to 1Mbp.

## Testing TE enrichments of zygotically activated genes (ZAGs)

To test whether TEs depleted in KO 2-cell embryos (adjusted P-value less than 0.05) are enriched in the neighbourhood of ZAGs, we counted the mean number of depleted TEs within ±10 kb of the transcription start site (TSS) of ZAGs (n = 698). ZAGs are defined as the WT class EU, which are differentially expressed in KO samples in the time-series cluster (*Figure 1—source data 4*). This process was then repeated 10,000 times for a randomly selected group of genes (n = 698). We excluded all depleted TEs overlapped with any exons. The observed mean number was then tested against a null model assuming no enrichment of depleted repeats in the neighbourhood of ZAGs by computing the empirical P-value of the observed mean number based on a null distribution of simulated mean numbers of depleted TEs (*Figure 4B*).

## Chimeric transcript identification

We extracted split (split but mapped to the same chromosome) or translocated (mapped to different chromosomes) reads from the remapped SAM files of GSNAP with a novel splicing option (–novelsplicing=1 –distant-splice-penalty=0). Only one genomic alignment was kept for each multi-mapped read. From the CIGAR string of the extracted reads, we calculated the genomic positions of both ends of split or translocated reads. If one end of a read is overlapped with one of genes and the other end is overlapped with one of repeat elements unambiguously, we considered it as the read supporting chimeric transcripts between genes and TE. Chimeric transcripts were validated through PCR amplification across the predicted chimeric junction and the DNA sequence at the junction was validated with Sanger sequencing.

## siRNA target analysis

To search full-length MuERV-L copies with the complete *gag* and *pol* genes, we aligned the sequences for MuERV-L *gag* and *pol* genes (GenBank accession number: Y12713) to the *Mus musculus* genome (GRCm38) using Bowtie2 (version 2.3.0) with options '-a -D 20 -R 3 -N 1 -L 20 –i S,1,0.50'. From the GTF file of RepeatMasker we used for TE counts, we extracted all MERVL-int elements overlapped with the alignments to the *gag* and *pol* genes (n = 583). To calculate the specificity of MuERV-L siRNA, we mapped the sequence of MuERV-L siRNA to the *Mus musculus* genome (GRCm38) using Bowtie2 (version 2.3.0) with options '-a -N 1 -L 10'. We stratified the SAM alignments into perfect match, one mismatch, and two mismatches by examining the 'XM:i:<N>' field and counted the alignments overlapped with the full-length MERVL-int elements.

## Confocal acquisition and image analysis

Full projections of images were taken every 0.5 μM (*Figure 5E*) or 1.5 μM (*Figure 3G*, *Figure 3—figure supplement 1C*, *Figure 5D and G*) along the z-axis with an inverted Leica TCS SP5 confocal microscope. For MuERV-L staining in 2-cell embryos, image analysis was carried out using Fiji

(*Schindelin et al., 2012*). Measurements of fluorescent intensity were automated using a custom ImageJ macro to create a Huang thresholded 3D cell mask from the fluorescent signal and measure the mean intensity inside each embryo. For measurements of the number of DAPI[+] cells in day five blastocysts (*Figure 5E*), we generated a Jython script. To allow reliable counting of clustered cells with widely varying amounts of labelling, a hierarchical k-means segmentation algorithm (*Dufour et al., 2008*) was implemented to generate a 3D mask, and the ImageJ 3D Object Counter plugin (*Bolte and Cordelières, 2006*) was used to determine how many separated objects were present by binary connexity analysis.

## Acknowledgements

We thank Richard Butler for his support on the confocal imaging analysis, Charles Bradshaw for bio-informatic support, Todd S Macfarlan and Samuel L Pfaff for the 2C::tdTomato ESCs. We also thank members of the Surani lab for their critical input and helpful discussions on this project. The work was funded by a studentship to YH from the James Baird Fund, University of Cambridge, by the DGIST Start-up Fund of the Ministry of Science, ICT and Future Planning to JKK, by a core grant from EMBL and CRUK to JCM, by a Wellcome Trust Senior Investigator Award to MAS, and by a core grant from the Wellcome Trust and Cancer Research UK to the Gurdon Institute.

## Additional information

### Funding

| Funder | Grant reference number | Author |
| --- | --- | --- |
| Wellcome | 096738 | Yun Huang<br>Dang Vinh Do<br>Caroline Lee<br>Christopher A Penfold<br>Jan J Zylicz<br>Jamie A Hackett<br>M Azim Surani |
| Wellcome | 092096 | Yun Huang<br>Dang Vinh Do<br>Caroline Lee<br>Christopher A Penfold<br>Jan J Zylicz<br>Jamie A Hackett<br>M Azim Surani |
| Cancer Research UK | C6946/A14492 | Yun Huang<br>Dang Vinh Do<br>Caroline Lee<br>Christopher A Penfold<br>Jan J Zylicz<br>Jamie A Hackett<br>M Azim Surani |
| James Baird Fund, University of Cambridge | | Yun Huang |
| European Molecular Biology Laboratory | | Jong Kyoung Kim<br>John C Marioni<br>Jamie A Hackett |
| Cancer Research UK | | Jong Kyoung Kim<br>John C Marioni |
| DGIST Start-up Fund of the Ministry of Science, ICT and Future Planning | 2017010073 | Jong Kyoung Kim |

The funders had no role in study design, data collection and interpretation, or the decision to submit the work for publication.

## Author contributions

YH, Conceptualization, Resources, Data curation, Formal analysis, Validation, Investigation, Methodology, Writing—original draft, Writing—review and editing; JKK, Resources, Data curation, Formal analysis, Investigation, Methodology, Writing—original draft, Writing—review and editing; DVD, CL, Data curation, Investigation, Methodology; CAP, Formal analysis, Investigation, Methodology; JJZ, Writing—review and editing, Critical evaluation of the manuscript for important intellectual content; JCM, Funding acquisition, Project administration, Writing—review and editing; JAH, Conceptualization, Supervision, Writing—original draft, Writing—review and editing; MAS, Conceptualization, Supervision, Funding acquisition, Writing—original draft, Project administration, Writing—review and editing

## Author ORCIDs

Yun Huang, http://orcid.org/0000-0001-7843-9126
Jan J Zylicz, http://orcid.org/0000-0001-9622-5658
Jamie A Hackett, http://orcid.org/0000-0002-6237-3684
M Azim Surani, http://orcid.org/0000-0002-8640-4318

## Ethics

Animal experimentation: All husbandry and experiments involving mice were authorized by a UK Home Office Project License 80/2637 and carried out in a Home Office-designated facility.

# Additional files

## Supplementary files

• Supplementary file 1. Sample details for single-cell / embryo RNA sequencing.

• Supplementary file 2. Stella ChIP-seq enriched peaks.

• Supplementary file 3. A list of primer sequences used in this study.

## Major datasets

The following dataset was generated:

| Author(s) | Year | Dataset title | Dataset URL | Database, license, and accessibility information |
|---|---|---|---|---|
| Huang Y, Kim JK, Do DV, Lee C, Penfold CA, Zylicz JJ, Marioni JC, Hackett JA, Surani MA | 2016 | Transcription profiling of wild type and Stella knockout oocytes, wild type and Stella maternal/zygotic knockout embryos to study to the role of Stella in early mouse development | http://www.ebi.ac.uk/arrayexpress/experiments/E-MTAB-5210/ | Publicly available at the ArrayExpress (accession no: E-MTAB-5210) |

The following previously published datasets were used:

| Author(s) | Year | Dataset title | Dataset URL | Database, license, and accessibility information |
|---|---|---|---|---|
| Macfarlan TS, Gifford WD, Driscoll S, Lettieri K, Rowe HM, Bonanomi D, Firth A, Singer O, Trono D, Pfaff SL. | 2012 | 2C::tomato ES cells, 2-cell embryos and wild type oocytes | https://www.ncbi.nlm.nih.gov/geo/query/acc.cgi?acc=GSE33923 | Publicly available at the NCBI Gene Expression Omnibus (accession no: GSE33923) |

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
