## [Decision Letter]

Thank you for submitting your article "STELLA and MuERV-L activation are essential for early mouse development" for consideration by *eLife*. Your article has been reviewed by three peer reviewers, and the evaluation has been overseen by Robb Krumlauf as the Reviewing Editor and Fiona Watt as the Senior Editor. The following individuals involved in review of your submission have agreed to reveal their identity: Didier Trono (Reviewer #1) and Todd S Macfarlan (Reviewer #2).

The reviewers have discussed the reviews with one another and the Reviewing Editor has drafted this decision to help you prepare a revised submission.

This study presents an interesting set of findings. The strength of this paper is in the genomic analyses, primarily single cell RNA-sequencing approaches and data analysis. They convincingly show failure in MZT of maternal Stella mutants, which is a significant and important finding, although not entirely surprising, since embryogenesis has been previously shown to be impaired in these mutants. There are however two major weaknesses to the work. The first shortcoming is a failure to provide mechanistic insight on how Stella might affect MERVL transcription and regulates MZT genes. The second relates to the lack of clarity of the proposed connection between Stella, MERVL, 2-cell stage-specific genes and MZT. The loss-of-function experiments (via RNAi) of MERVL elements is not entirely convincing. The knockdown as performed should not affect the expression of MuERVL LTR-driven ZGA genes, which are also controlled by STELLA. This offers no mechanistic insight into how the expression of MERVL elements is important for developmental progression. Thus more mechanistic insights would greatly improve the manuscript. However, we realize that extra work showing the requirement of MuERV-L for development might take an extended period that would prevent a timely submission. The authors could change to this conclusion and instead focus on what is regulated by Stella. Hence, as a minimum, there is a need for an enhanced discussion with plausible mechanisms of both Stella and MERVLs role in early development. Furthermore there is a need for controls to address concerns about the interpretation of the MERVL siRNA experiment tests.

Major specific points:

1) Mechanism of Stella action. The authors don't attempt any experiment to address how Stella regulates MZT, other than looking at effects on transcription. No data is shown to demonstrate that these effects are direct. It is unlikely that Stella binds both to maternal mRNAs aiding in their destruction while simultaneously activation MERVL promoters like a transcription factor. As a starting point, it would be interesting if the authors could "rescue" the maternal mutants by injecting Stella cDNAs containing mutations in some of its putative domains to tease out domains that might be important for the observed MZT defects. Likewise, the Introduction would benefit from a description of the Stella protein and its known domains.

2) MERVL loss-of-function experiments. The authors use a siRNA against the MEVL GAG region, which will target some but not all full-length MERVL elements. The authors should provide a list of the genomic MERVL elements (or the percentage) that should be perfect targets, one mismatch, two mismatch, etc. to the siRNA. In other words, provide the reader with the fraction of full-length MERVL elements that should be affected by the siRNA. Additionally, the IF images, even if very carefully performed, can be easily misinterpreted since the embryos are imaged separately. Western blots with the same Gag antibody should also be included. This is achievable since Heidmann's lab reported westerns from 2C embryos over a decade ago.

3) In Figure 5, it looks like the scramble siRNA reduces MERVL expression relative to uninjected embryos. Is this significant? In Figure 5, the authors should include uninjected embryos for comparison and include statistics. There is a concern that the different phenotypes between scramble siRNA and MERVL siRNA are much less than the difference between uninjected and scramble siRNA. Thus, the reader should be provided this control. Figure 5 . Given the title, replicating this a few more times to get higher certainty in the effect of MuERV-L knockdown would be appropriate. If this work would be substantial, a change in title and claims about development potential needs to be considered.

4) There is no mechanistic insight into how knocking down MERVL transcripts (and Gag positive virus particles) affects development and the discussion on this topic is not adequate. In the Discussion, the authors state they found "that reducing MERVL activation at the 2C stage affects development", but this is misleading, since knocking down MERVL transcripts with an siRNA is not the same as preventing the expression of MERVL elements and the broad open chromatin domains this likely causes. This is testable and one might expect that MERVL siRNAs have little effect on the embryonic transcriptome or chromatin state surrounding MERVLs. A better experiment with this aim would be CRISPRi. It is nonetheless interesting that MERVL element mRNA/proteins seem to be important for development, but there is no experiment that addresses mechanism, or even discussion on a plausible mechanism. For example, might the viral particles themselves be important, similar to the roles proposed for HERVK particles in human embryos (Grow et al., 2015). Since this is a major claim of the paper, it either needs to be softened or it should be supported with more compelling data such as rescue experiments.

5) It would be interesting to test if the MERVL LOF phenotype is more severe if a portion of the M2t_mm LTR is targeted by an siRNA, since this could potentially target chimeric transcripts in addition to the full-length elements. Was this experiment attempted?

6) Figure 1 nicely demonstrates that a subset of MZT genes fails to become activated in Stella KO embryos. The GO analysis illustrated in Figure 2 further reveals that a lot of different pathways are affected, although none stands out as particularly responsible for the observed phenotype. Panels in 2C and 2D provide useful validations of the RNA seq analyses, yet do not add any specific information hence could be presented as Supplementary Data.

7) Figure 3 depicts the dysregulation of TEs expression in Stella KO embryos. How many cells are included for each stage in Figure 3 should be indicated. The authors should verify levels of MERVL expression by qPCR, using consensus primers spanning the sequence of this element. They further suggest that a zinc finger protein, ZFP143, might be involved in regulating MERVL, as its levels increase slightly upon passage from the 1-cell to the 2-cell stage, and are higher in the absence of Stella. The upper graph needs more information: how many cells assessed? Statistical significance? Normalization? Furthermore, there is no reference regarding the putative ZFP143-binding sequence, the location of which within MERVL (and other TEs observed to be upregulated in the absence of Stella) should be indicated. As well, to support their incrimination of ZFP153 in the control of MERVL, the authors could overexpress it or knock it down in mESC, a small fraction of which upregulates MERVL at any given time in tissue culture (McFarlan et al., 2012). As well, the effect of Stella and ZFP143 overexpression on MERVL LTR-driven reporters could be tested in this or other heterologous systems.

8) In line with the decreased upregulation of MERVL elements and their MT2_mm LTRs in the absence of Stella, Figure 4 reveals that the expression of genes driven from these transcriptional units is also reduced in this setting, suggesting that Stella regulates the transcriptional activity of the MERVL promoter. In Figure 5 the authors demonstrate that depletion by siRNA of MERVL transcripts from early embryos leads to a phenotype closely similar to that induced by the depletion of Stella. However, the siRNA used in these experiments targets the gag sequence of MERVL, hence should trigger the degradation of only a subset of MERVL RNAs, and not those where the LTR drives chimeric transcripts, and drives MZT genes, as those illustrated in Figure 4. It might be interpreted as suggesting that the lethal phenotype induces by Stella depletion results from the lack of MERVL transcripts, not from the modulation of MZT genes. The authors should definitely address this paradox, for instance by i) quantifying the expression of MZT genes (as per Figure 4) upon MERVL depletion, and ii) injecting MERVL transcripts in the zygote of Stella KO embryos to see whether they rescue lethality.

9) Figure 4 illustrates 4 examples of genes, the expression trend of which matches that of neighboring TEs. In only one case (ZFP54) is this TE a MERVL-related element, whereas in the other 3 they are other types of TEs. The nature of these TEs should be better described (LTRs? Internal sequences? Orientation?). Were chimeric transcripts detected in any of these cases? Were there other TEs in the intervening sequences (between TEs mentioned here and genes and their TSS -genes orientation should also be indicated).

[Editors' note: further revisions were requested prior to acceptance, as described below.]

Thank you for resubmitting your work entitled "STELLA modulates transcriptional and endogenous retrovirus programs during maternal-to-zygotic transition" for further consideration at *eLife*. Your revised article has been favorably evaluated by Fiona Watt (Senior editor), a Reviewing editor, and three reviewers.

The manuscript has been improved but there are some remaining issues that need to be addressed before acceptance, as outlined below by reviewer 1. The major issues that need to be addressed relate to conclusions of MERVL depletion in early embryos reducing level of 2C specific genes and the relevance of Figure 5—figure supplement 5A.

*Reviewer #1:*

Most of the concerns we raised in our previous review have been satisfactorily addressed in the new version of the manuscript. However, the experiments aimed at exposing the importance and role of MERVL transcripts in MZT still raise questions, and need further clarification or dampening of the related claims.

In the manuscript and rebuttal letter, the authors state that MERVL depletion in early embryos leads to decreased expression of 2C-specific genes, but the data do not fully support this claim. Figure 5—figure supplement 1 shows qRT-PCR analysis for 4 2C-specific genes driven by the MT2_mm MERVL LTR in embryos injected with MERVL-targeting or scramble siRNA. According to the authors' model, they should be downregulated upon knocking down MERVL transcripts. Yet two (Tcstv1, Gm4340) are upregulated, one (Gpbp1l1) unchanged, and only one (Sp110) downregulated, similar to the targeted MERVL transcripts themselves. It would have been useful to examine the siRNA-sensitivity of some other of the 200 or so MZT genes, including ones not driven by a MERVL-associated LTR but either by the LTR of another transposable element (e.g. Eif1a-like genes) or their own promoter (e.g. Zscan4).

As well, the experiment described in Figure 5—figure supplement 5A (could not find where it is specifically described in the text) does not add much, since the downregulation from the injection of an siRNA targeting the MT2_mm part of chimeric transcripts is significant for only 2 out of 6 of tested here.

---

## [Author Response]

*Major specific points:*

*1) Mechanism of Stella action. The authors don't attempt any experiment to address how Stella regulates MZT, other than looking at effects on transcription. No data is shown to demonstrate that these effects are direct. It is unlikely that Stella binds both to maternal mRNAs aiding in their destruction while simultaneously activation MERVL promoters like a transcription factor.*

Our aim at the very outset was to understand the mechanism of Stella function, while accepting that many of the experiments (e.g. ChIP, Co-IP, CLIP etc) are very challenging in 2-cell embryos. We have therefore attempted to identify direct targets of Stella in mESCs, using a published protocol with modifications (Nakamura et al., 2012) (Figure 6). We tagged hemagglutinin (HA) to the C-terminus of Stella and used the high affinity HA antibody to pull down Stella bound chromatin. The Stella+HA construct was over-expressed in the absence of endogenous Stella by using StellaM/Z KO ESCs, with NANOG-HA and eGFP-HA as positive and negative controls, respectively; expression was confirmed by western blot (Figure 6).

Author response image 1.Stella-HA ChIP.(**A**) Western blot of Stella and HA. Stella with 3 copies of hemagglutinin (HA) tagged to its C-terminal is over-expressed in a Dox inducible system in StellaM/Z KO ESCs. Stella’s predicted size is ~ 21kDA, and appears as ~ 27kDA due to addition of 3xHA construct. Bottom panel shows H3 loading control. (**B**) Western blot for NANOG+HA and eGFP+HA. Both NANOG+HA and eGFP+HA constructs were expressed in a Dox-inducible system in StellaM/Z KO ESCs. The blot show NANOG+HA construct is detected at ~55kDA while the endogenous NANOG is detected at ~45kDA. eGFP+HA construct is detected at ~37kDA, as confirmed with eGFP antibody alone. Bottom panel shows H3 loading control. (**C**) Top panel shows ChIP-qPCR of the relative enrichment of NANOG-HA in NANOG known binding regions (Xist, proximal enhancer of Prdm14) and gene desert regions (GAPDH and Chr18). Bottom panel shows ChIP-qPCR of the relative enrichment of Stella-HA in previously published Stella binding regions (Mage-a2, Wfdc15a) (Nakamura et al., 2012) and gene desert regions. One-tailed Student’s T-test was performed, where * corresponds to p<0.05 or otherwise stated.**DOI:**
http://dx.doi.org/10.7554/eLife.22345.027

We detected enrichments of NANOG in known binding sites (Xist, proximal enhancers of Prdm14) (Gillich, Bao, and Surani 2013, Murakami et al., 2016), but not for eGFP, supporting the efficacy of the ChIP. However, there was little or no enrichment of Stella at its previously reported binding sites (Mage-a2, Wfdc15a1) (Nakamura et al., 2012), nor could we detect Stella binding to other regions in the genome (n=56) (subsection “Stella is associated with the expression of specific TE “, [Supplementary-material SD10-data]). In addition, neither over-expression nor knockdown of *Dppa3* affected MuERV-L expression in mESCs (Figure 3—figure supplement 2). This indicates that Stella does not bind to zygotic genes / MuERV-L elements in mESCs, with the caveat that mESC might not accurately reflect the cellular/chromatin context of 2-cell embryos, where these experiments are intractable. Nonetheless, we felt the attempt was worthwhile partly because published reports indicated that ESCs cycle through a 2C-like state (Macfarlan et al., 2012).

*As a starting point, it would be interesting if the authors could "rescue" the maternal mutants by injecting Stella cDNAs containing mutations in some of its putative domains to tease out domains that might be important for the observed MZT defects.*

We considered this point previously by comparing maternal- and zygotic- only mutants. Briefly, maternal Stella persists until ~E1.5, followed by zygotic Stella expression at ~E2.5 embryos (Payer et al. 2003). Evidence suggests that maternal Stella is critical for development, but zygotic Stella is largely dispensable, implying that reintroduction of Stella in the early embryo cannot rescue the effects of maternal loss (Payer et al., 2003, Bortvin et al., 2004, Nakamura et al., 2007). This emphasizes the critical role of Stella at the earliest developmental stages. Thus, it is doubtful that attempts to conduct ‘rescue’ experiments would be effective since Stella protein is unlikely to accumulate to sufficient levels to have an effect. Note also that these experiments are very challenging since this would require a large number of mutant oocytes.

*Likewise, the Introduction would benefit from a description of the Stella protein and its known domains.*

We provide the information together with a schematic diagram in the Introduction (Figure 1—figure supplement 1).

*2) MERVL loss-of-function experiments. The authors use a siRNA against the MEVL GAG region, which will target some but not all full-length MERVL elements. The authors should provide a list of the genomic MERVL elements (or the percentage) that should be perfect targets, one mismatch, two mismatch, etc. to the siRNA. In other words, provide the reader with the fraction of full-length MERVL elements that should be affected by the siRNA.*

This is an important point that we had overlooked. We have now performed the additional bioinformatics analysis (Figure 5) as suggested by the reviewers. We identified 583 full-length MuERV-L elements that fully encode the *gag* and *pol* proteins in the genome. MuERV-L siRNA targets 80.5% of full-length MuERV-L elements with perfect match and 99.5% of full-length MuERV-L elements with 2 mismatches or less. This is an informative result given the minimum number of predicted off-targets with 2 mismatches or less. Furthermore, scramble siRNA did not have any predicted targets with 2 mismatches or less.

*Additionally, the IF images, even if very carefully performed, can be easily misinterpreted since the embryos are imaged separately. Western blots with the same Gag antibody should also be included. This is achievable since Heidmann's lab reported westerns from 2C embryos over a decade ago.*

While we acknowledge this point, the western blots would require approximately 40 – 100 2-cell embryos per sample (Kang et al., 2014, Komatsu et al., 2014, Mu et al., 2011). Note also that the experiments require microinjections, with a ~50% survival rate. Finally our MuERV-L antibody is different from that used by Heidmann’s lab (Ribet et al., 2008), and we have no access to it. For these reasons, we chose IF to validate MuERV-L protein knockdown. The analysis was carried with care using our outstanding core imaging facility at the Gurdon Institute, with high technical expertise. Our facility has designed a custom ImageJ macro plugin to measure IF signals; we used this to asses MuERV-L (GFP) intensity in 2-cell embryos using the Fiji image software.

Accordingly, we took extra precautions ensuring that all the control and experimental samples were treated simultaneously and identically, and on the same day with identical microscope settings; representative images for each condition are shown in Figure 5. The quantification of the GFP intensity is however meant to provide changes in relative levels in MuERV-L protein, and not precise levels of the protein. This procedure for IF images was used recently by our imaging core on another study on pre-implantation mouse embryos (Goolam et al.Cell, 2016).

*3) In Figure 5, it looks like the scramble siRNA reduces MERVL expression relative to uninjected embryos. Is this significant?*

MuERV-L pol expression is not significantly reduced in scramble siRNA compared to uninjected embryos as judged by the two-sided Wilcoxson rank sum test and this is now labeled as n.s. (non significant) in Figure 5 and Figure 5 for clarity.

*In Figure 5, the authors should include uninjected embryos for comparison and include statistics. There is a concern that the different phenotypes between scramble siRNA and MERVL siRNA are much less than the difference between uninjected and scramble siRNA. Thus, the reader should be provided this control. Figure 5 .*

The uninjected control embryos are now included in the Figure 5 and Figure 5 as suggested by the reviewers. At 20μM, the percentage of embryos reaching blastocyst by day 4 was similar between uninjected and scramble siRNA embryos (83% versus 89% respectively). However, only 60% of MuERV-L siRNA injected embryos reached the blastocyst stage, which narrowly below significance (p=0.065). There were no other statistically significant differences between uninjected, scramble or MuERV-L siRNA treated embryos in Figure 5. Statistical significance is now stated for Figure 5.

We agree that at 80μM concentration, siRNA itself may have a negative effect on embryo development compared to uninjected embryos. This might in part be due to some non-specific effects, which we acknowledged in the manuscript (subsection “MuERV-L plays a functional role during pre-implantation development “). On the other hand, there are 583 full-length copies of MuERV-L elements in the genome, which are highly upregulated at the 2-cell stage, comprising ~3% of the transcriprome. This likely requires a higher concentration of siRNA to achieve a significant knockdown compared to what is normally the case for a single copy gene. To accommodate the conflicting experimental drawbacks, we decided to present both 20μM and 80μM; they both show a similar effect, with the latter having a greater effect accepting the appropriate caveats.

*Given the title, replicating this a few more times to get higher certainty in the effect of MuERV-L knockdown would be appropriate. If this work would be substantial, a change in title and claims about development potential needs to be considered.*

We agree with the reviewers’ comments and appreciate the suggestion offered by the reviewers. Obtaining enough mutant embryos is very challenging besides the experiments themselves. We have therefore changed the title and moderated the claims in the manuscript. The revised title is below, which we hope will be acceptable.

“Stella modulates transcriptional and endogenous retrovirus programs during maternal-to-zygotic transition”

*4) There is no mechanistic insight into how knocking down MERVL transcripts (and Gag positive virus particles) affects development and the discussion on this topic is not adequate. In the Discussion, the authors state they found "that reducing MERVL activation at the 2C stage affects development", but this is misleading, since knocking down MERVL transcripts with an siRNA is not the same as preventing the expression of MERVL elements and the broad open chromatin domains this likely causes. This is testable and one might expect that MERVL siRNAs have little effect on the embryonic transcriptome or chromatin state surrounding MERVLs. A better experiment with this aim would be CRISPRi. It is nonetheless interesting that MERVL element mRNA/proteins seem to be important for development, but there is no experiment that addresses mechanism, or even discussion on a plausible mechanism. For example, might the viral particles themselves be important, similar to the roles proposed for HERVK particles in human embryos (Grow et al., Nature). Since this is a major claim of the paper, it either needs to be softened or it should be supported with more compelling data such as rescue experiments.*

As we pointed out earlier, we have moderated our claims based on the reviewers’ comments, with respect to the siRNA effects on MuERV-L transcript and protein. Also in response to points 4 and point 8, we have performed additional qRT-PCR analysis on 2-cell embryos, which revealed that MuERV-L knockdown does affect the expression of chimeric transcripts and 2-cell specific genes, which may contribute to the reduced embryonic potential of MuERV-L knockdown embryos. Importantly, this suggests siRNA targeting is one relevant approach to investigate the biological significance of gene expression derived from MuERV-L.

The use of CRISPRi is a potentially interesting and complementary approach to address whether LTR-driven zygotic gene expression is important during development. However, it will be difficult to generate a gRNA library capable of targeting most of the LTR promoters without significant indirect effects on protein-coding genes, as many solo or non-functional LTRs have integrated into 5’ and 3’UTRs, which would be targeted despite little or no influence on the expression of their host gene. Thus developing this CRISPRi system would require extensive optimization, including efficient delivery into the zygote.

A potential role of viral particles in mouse development seems less likely, because we previously showed that knockout of all interferon response genes (IFITM) family members has no detectable effect on mouse development (Lange et al., 2008). Furthermore, while HERVK and virus-like particles were indicated in human hypomethylated epiblast cells in blastocysts (Grow et al., 2015), we did not detect them in highly hypomethylated human primordial germ cells that were predicted in the paper by Grow (W.Tang et al., unpublished). We have included discussion of this and additional possible mechanisms by which MuERV-L mRNA / protein may function during pre-implantation development (Discussion section paragraph five).

*5) It would be interesting to test if the MERVL LOF phenotype is more severe if a portion of the M2t_mm LTR is targeted by an siRNA, since this could potentially target chimeric transcripts in addition to the full-length elements. Was this experiment attempted?*

We have preliminary data regarding this specific comment (Figure 7). We designed siRNA against MT2_Mm and MT2 elements (MT2A, MT2B, MT2B1, MT2B2, MT2C, MT2C_Mm); these LTRs were also significantly depleted in Stella M/Z KO 2-cell embryos. The phenotype we observed for MT2_Mm knockdown was in fact similar to MuERV-L knockdown (Figure 7). Bioinformatics analysis predicted MT2_Mm siRNA targets 74.3% of MT2_Mm elements with 2 or less mismatches and a minimum number of off-targets (Figure 7). MT2 elements have very diverse sequences, with 37002 elements considered in the analysis, and only 7.85% of MT2 elements targeted by MT2 siRNA with 2 or less mismatches (Figure 7). Nevertheless, we observed an effect on embryonic development (Figure 7), which may have resulted from knockdown of a combination of MT2 elements and associated chimeric transcripts.

Author response image 2.Effects of MT2 element knockdown on pre-implantation embryonic development.(**A**) A bar graph illustrating the number of uninjected embryos, embryos injected with 20μM scramble or 20μM MT2_Mm siRNAs, at different stages of development in day 2 – 4 of *in-vitro* culture, n=1. The percentage of embryos reaching the blastocyst stage is stated. (**B**) Bioinformatics analysis of the number of MT2_Mm elements and off-targets that are perfect matches or 1 or 2 mismatches for MT2_Mm siRNA. A total of 2667 MT2_Mm elements are included in the analysis. (**C**) A bar graph illustrating the number of uninjected embryos, embryos injected with 20 μM scramble or 20 μM MT2 siRNAs, at different stages of development in day 2 – 4 of *in-vitro* culture, n=1. The percentage of embryos reaching the blastocyst stage is stated. (**D**) Bioinformatics analysis of the number of MT2 elements (MT2A, MT2B, MT2B1, MT2B2, MT2C_Mm) and off-targets that are perfect matches or 1 or 2 mismatches for MT2 siRNA. A total of 37002 MT2 elements are included in the analysis.**DOI:**
http://dx.doi.org/10.7554/eLife.22345.028

*6) Figure 1 nicely demonstrates that a subset of MZT genes fails to become activated in Stella KO embryos. The GO analysis illustrated in Figure 2 further reveals that a lot of different pathways are affected, although none stands out as particularly responsible for the observed phenotype. Panels in 2C and 2D provide useful validations of the RNA seq analyses, yet do not add any specific information hence could be presented as Supplementary Data.*

There were significant dysregulations observed in Stella M/Z KO 2-cell embryos and the GO analysis served to highlight potentially significant functional pathways that contributed towards embryonic lethality in KO embryos. We believe the qRT-PCR data give examples from each of the pathways, provides validation of the RNA-seq from independent experiments (in an alternative visual format) and further emphasizes the potentially important biological processes misregulated in Stella M/Z KO 2-cell embryos. Thus we would prefer to retain it in the main figure if possible.

*7) Figure 3 depicts the dysregulation of TEs expression in Stella KO embryos. How many cells are included for each stage in Figure 3 should be indicated.*

The bar chart for Figure 3 were calculated from single cell / embryo RNA sequencing reads, which were mapped to TEs. We have now made this clear in the figure legend. Thus the number of embryos included for each stage is the same as the original RNA-sequencing study (WT oocyte n=12, KO oocyte n=12, WT 1-cell n=12, KO 1-cell n=12, WT 2-cell n=9, KO 2-cell n=9).

*The authors should verify levels of MERVL expression by qPCR, using consensus primers spanning the sequence of this element.*

To explore this we quantified MuERV-L expression in WT and KO 2-cell embryos processed for RNA-sequencing (Figure 8). RNA-seq mapped reads showed significant reduction in MuERV-L-int expression in KO 2-cell embryos. This was not however observed in 2 sets of MuERV-L by qRT-PCR analysis (Figure 8). Initial evidence suggests this is because MuERV-L primers can amplify significant off target regions, which lead to over-estimation of MuERV-L expression in some embryos (Figure 8). For this reason we opted for IF validation, which showed robust reduction of MuERV-L Gag protein level in KO 2-cell embryos, and corroborated the unbiased RNA-seq, which showed a comparable reduction.

Author response image 3.MERVL RNA-sequencing versus qRT-PCR.(**A**) The first dot plot shows MERVL-Int expression in WT and KO 2-cell embryos. MERVL-Int reads were mapped from single embryo RNA-sequencing and expressed as counts per million. The latter two dot plots shows MERVL expression from 2 qRT-PCR primers (MERVL-pol and MERVL1) in the same WT and KO 2-cell embryos. Samples are expressed as a ratio relative to WT embryo 1 (WT1). The median and interquartile range is indicated. WT n=9, KO n=9. Two-sided Wilcoxon rank sum test was performed and *** corresponds to p<0.001, n.s = not significant. (**B**) A plot of expression correlation between MERVL-Int from RNA-sequencing and MERVL-pol from qRT-PCR, Pearson correlation analysis performed.**DOI:**
http://dx.doi.org/10.7554/eLife.22345.029

*They further suggest that a zinc finger protein, ZFP143, might be involved in regulating MERVL, as its levels increase slightly upon passage from the 1-cell to the 2-cell stage, and are higher in the absence of Stella. The upper graph needs more information: how many cells assessed? Statistical significance? Normalization? Furthermore, there is no reference regarding the putative ZFP143-binding sequence, the location of which within MERVL (and other TEs observed to be upregulated in the absence of Stella) should be indicated. As well, to support their incrimination of ZFP153 in the control of MERVL, the authors could overexpress it or knock it down in mESC, a small fraction of which upregulates MERVL at any given time in tissue culture (McFarlan et al., 2012). As well, the effect of Stella and ZFP143 overexpression on MERVL LTR-driven reporters could be tested in this or other heterologous systems.*

Thank you for your comments and interest on ZFP143 as a potential candidate regulating MuERV-L elements. We illustrated ZFP143 as an example of a motif enriched in differentially expressed TEs between WT and KO samples, but this is not the main focus here. As we do not have sufficient experimental evidence or bioinformatics analysis to suggest ZFP143 directly regulates TEs / MuERV-L elements, we have removed this from the manuscript.

*8) In line with the decreased upregulation of MERVL elements and their MT2_mm LTRs in the absence of Stella, Figure 4 reveals that the expression of genes driven from these transcriptional units is also reduced in this setting, suggesting that Stella regulates the transcriptional activity of the MERVL promoter. In Figure 5 the authors demonstrate that depletion by siRNA of MERVL transcripts from early embryos leads to a phenotype closely similar to that induced by the depletion of Stella. However, the siRNA used in these experiments targets the gag sequence of MERVL, hence should trigger the degradation of only a subset of MERVL RNAs, and not those where the LTR drives chimeric transcripts, and drives MZT genes, as those illustrated in Figure 4. It might be interpreted as suggesting that the lethal phenotype induces by Stella depletion results from the lack of MERVL transcripts, not from the modulation of MZT genes. The authors should definitely address this paradox, for instance by i) quantifying the expression of MZT genes (as per Figure 4) upon MERVL depletion, and ii) injecting MERVL transcripts in the zygote of Stella KO embryos to see whether they rescue lethality.*

Thank you for pointing out the contradictory messages in the manuscript. We have attempted to address this as suggested by the reviewers (i), by quantifying transcript expression in uninjected, 80 μM scramble and 80 μM MuERV-L siRNA 2-cell embryos (Figure 5—figure supplement 1). Interestingly, we observed MuERV-L knockdown results in significantly reduced expression of 2/6 of chimeric transcripts (*Rbm8a, GM12617*), and in a 2-cell specific gene – *Sp110*. Furthermore, *GM12617* is also significantly depleted in StellaM/Z KO 2-cell embryos ([Supplementary-material SD2-data]). This suggests that defects in transcript expression may partially contribute to the reduced developmental potential of MuERV-L knockdown embryos.

In response to the second experiment proposed (ii), while MuERV-L transcripts are significantly depleted in StellaM/Z KO 2-cell embryos, we have also identified 2560 misregulated genes. We believe the aberrant ERV programme (including MuERV-L) may have contributed towards gene misregulation, however, MuERV-L transcript alone will likely not be sufficient to rescue the embryonic lethality observed in StellaM/Z KO embryos.

*9) Figure 4 illustrates 4 examples of genes, the expression trend of which matches that of neighboring TEs. In only one case (ZFP54) is this TE a MERVL-related element, whereas in the other 3 they are other types of TEs. The nature of these TEs should be better described (LTRs? Internal sequences? Orientation?). Were chimeric transcripts detected in any of these cases? Were there other TEs in the intervening sequences (between TEs mentioned here and genes and their TSS -genes orientation should also be indicated).*

The nature of the 4 TE types are mentioned in the manuscript (subsection “Expression of a subset of TEs is positively correlated with their nearest gene “) and the orientations of the TEs and genes are illustrated in Figure 4 now. ZFP54-MT2B is a chimeric transcript (Macfarlan et al., 2012), but the other 3 pairs of TE/gene are not. These candidates were chosen based on a genome-wide expression correlation analysis between a TE and its nearest gene. While there may be other TEs in the intervening sequences between some pairs of TE/gene, these intervening TEs are not robustly expressed in 2-cell embryos.

[Editors' note: further revisions were requested prior to acceptance, as described below.]

*Reviewer #1:*

*Most of the concerns we raised in our previous review have been satisfactorily addressed in the new version of the manuscript. However, the experiments aimed at exposing the importance and role of MERVL transcripts in MZT still raise questions, and need further clarification or dampening of the related claims.*

*In the manuscript and rebuttal letter, the authors state that MERVL depletion in early embryos leads to decreased expression of 2C-specific genes, but the data do not fully support this claim. Figure 5—figure supplement 1 shows qRT-PCR analysis for 4 2C-specific genes driven by the MT2_mm MERVL LTR in embryos injected with MERVL-targeting or scramble siRNA. According to the authors' model, they should be downregulated upon knocking down MERVL transcripts. Yet two (Tcstv1, Gm4340) are upregulated, one (Gpbp1l1) unchanged, and only one (Sp110) downregulated, similar to the targeted MERVL transcripts themselves. It would have been useful to examine the siRNA-sensitivity of some other of the 200 or so MZT genes, including ones not driven by a MERVL-associated LTR but either by the LTR of another transposable element (e.g. Eif1a-like genes) or their own promoter (e.g. Zscan4).*

As well, the experiment described in Figure 5—figure supplement 5A (could not find where it is specifically described in the text) does not add much, since the downregulation from the injection of an siRNA targeting the MT2_mm part of chimeric transcripts is significant for only 2 out of 6 of tested here.

Concerning MuERV-L KD gene expression analysis, we had attempted to characterize the effect on a small sub-set of 2C specific genes; this showed some variable effects. In light of this we have accepted the reviewer’s suggestion to dampen our claims. We have therefore decided to remove this section from the manuscript and Figure 5—figure supplement 1.

Our analysis of chimeric transcripts however revealed that all analysed amplicons tended towards reduced expression in MuERV-L knockdown embryos, with 2 transcripts being significantly depleted (*p<0.05*) (Figure 5—figure supplement 1). We believe that it will be valuable for the readers to see these observations since it indicates that MuERV-L depletion could affect embryonic development via multiple mechanisms.

We have however removed all the conclusions based on MuERV-L KD chimeric transcript expression analysis and simply present the observation in the Results section. We suggest in the discussion that one possible consequence of MuERV-L knockdown is that it may affect expression of chimeric transcripts, but we state that this warrants further investigation.